# Positional Encoding for Spiking Transformers

**Zijian Zhou**[1]  **Yu Liang**[1]  **Honglin Cao**[1]  **Ammar Belatreche**[2]  **Jieyuan Zhang**[1]  **Wenjie Wei**[1]  **Shuai Wang**[1]
**Malu Zhang**[1 3]  **Yang Yang**[1]  **Haizhou Li**[3 4]

## Abstract

Transformer-based Spiking Neural Networks (SNNs) have recently emerged as a promising paradigm to sequential modeling, combining the strong representational capabilities of Transformers with the sparse spike-driven computation of SNNs. Within such position-agnostic architectures, positional encoding is critical for injecting order information, allowing the model to distinguish token positions, capture sequential dependencies, and represent relative relationships among tokens. However, existing positional encoding methods for SNNs are largely inherited from ANNs and, in doing so, undermine the spike-driven computational properties that are central to spiking transformers. To address this limitation, we propose the Spiking Positional Encoding (SPE), a method designed specifically for Spiking Transformers, aimed at encoding relative positional information while preserving both spike-driven computation and the linear complexity of spiking self-attention. The core component of SPE is the Positional Encoding Leaky Integrate-and-Fire (PE-LIF) neuron, which incorporates position-dependent signals into neuronal thresholds and implicitly propagates this information through spike trains via continuous firing and membrane potential reset dynamics. Extensive experiments on thirteen NLP benchmarks demonstrate that SPE consistently outperforms existing SNN positional encoding methods, strengthens the sequence modeling capability, and improves energy efficiency without introducing additional trainable parameters. Code is available at https://github.com/CayleyZ/SPE.

[1]University of Electronic Science and Technology of China [2]Northumbria University [3]Shenzhen Loop Area Institute [4]The Chinese University of Hong Kong, Shenzhen (CUHK-Shenzhen). Correspondence to: Malu Zhang <maluzhang@uestc.edu.cn>.

*Proceedings of the 43rd International Conference on Machine Learning*, Seoul, South Korea. PMLR 306, 2026. Copyright 2026 by the author(s).

## 1. Introduction

Spiking Neural Networks (SNNs), characterized as third-generation artificial neural networks, have attracted considerable research interest due to their superior energy efficiency and enhanced biological plausibility compared to conventional Artificial Neural Networks (ANNs) (Maass, 1997). Unlike conventional ANNs that rely on continuous-valued activations, SNNs employ binary spikes as information carriers and operate within a temporally sparse, event-driven processing paradigm (Zhang et al., 2021; Tang et al., 2024). The inherently sparse synaptic communication in SNNs enables the replacement of computationally intensive multiply-accumulate (MAC) operations with more efficient accumulate (AC) operations, resulting in substantial improvements in computational efficiency (Wei et al., 2025; Liang et al., 2025; Liu et al., 2026). This energy-efficient property has driven the development of specialized neuromorphic hardware implementations, including SpiNNaker (Painkras et al., 2013), TrueNorth (Akopyan et al., 2015), Loihi (Davies et al., 2018), and Tianjic (Pei et al., 2019).

Despite these advantages, existing SNNs typically suffer from performance limitations that constrain their widespread application, particularly when compared to their ANN counterparts. To address these performance gaps while preserving the inherent benefits of spike-driven computation, recent years have witnessed several studies integrating Transformers into SNNs, leading to a series of high-performance models, such as Spikformer (Zhou et al., 2023), Spikingformer (Zhou et al., 2026), Spike-Driven Transformer v1, v2, v3 (Yao et al., 2023; 2024; 2025) and SpikeLM (Xing et al., 2024b). Compared to traditional convolutional architectures in SNNs, these Transformer-based models have demonstrated significant performance improvements (Zhang et al., 2025; Cao et al., 2025; Wei et al., 2026), highlighting their potential to integrate the computational efficiency of SNNs with the powerful expressive capability of Transformer architectures.

In sequential tasks such as Natural Language Processing (NLP), positional information is crucial. However, Transformer models suffer from an inherent limitation of being position-agnostic, thus lacking the capability to capture positional information. To address this issue, positional encod-

ing techniques have been proposed. While numerous positional encoding methods exist in the ANN domain (Vaswani et al., 2017; Shaw et al., 2018; Devlin et al., 2019; Lan et al., 2020; Su et al., 2024), these approaches cannot be directly applied to SNNs as they would disrupt the spike-driven paradigm. Positional encoding methods are primarily categorized into two types: absolute and relative positional encoding. While absolute positional encoding is simple to implement, it cannot represent relative positional information between different tokens, which is essential for capturing time-translational invariance property in sequential modeling. Relative positional encoding, on the other hand, captures relative positional information effectively but is typically incompatible with linear attention mechanisms. Therefore, it is necessary to develop a positional encoding method tailored for Spiking Transformers that can both represent relative positional information and integrate seamlessly with their linear attention mechanism.

In this paper, we first analyze why Spiking Transformers require positional encoding, then propose four design principles for positional encoding method tailored to Spiking Transformers. Guided by these principles, we introduce Spiking Positional Encoding (SPE), a method specifically designed for Spiking Transformers that can encode relative positional information. The core component of SPE is the Positional Encoding Leaky Integrate-and-Fire (PE-LIF) neuron layer, a variant of the soft-reset LIF neuron, which encodes positional information directly into neuron thresholds. This information is then reflected in the emitted spikes through continuous spike firing and membrane potential reset processes. We theoretically demonstrate that SPE possesses the capability to represent relative positional information and prove that SPE exhibits desirable long-term decay properties. Notably, SPE fully satisfies the four design principles we established, confirming the soundness of its design. We evaluate SPE on thirteen NLP datasets, consistently outperforming existing positional encoding methods.

- We present an analysis of the necessity of positional encoding in Spiking Transformers and establish four design principles that guide the development of effective positional encoding tailored for Spiking Transformers.

- We propose SPE, a novel positional encoding approach specifically designed for Spiking Transformers that captures relative positional information through the innovative PE-LIF neuron layer, which encodes positional information directly into neuron thresholds.

- We provide rigorous theoretical analysis demonstrating that SPE possesses the capability to represent relative positional information and prove that it exhibits desirable long-term decay properties, ensuring adherence to our proposed four design principles.

- We evaluate SPE extensively across thirteen NLP datasets, including GLUE and other widely-adopted benchmarks, demonstrating consistent improvements over existing positional encoding methods.

## 2. Related Works

### 2.1. Spiking Transformers

Spiking Transformers integrate SNNs with Transformer architectures, combining biological plausibility and energy efficiency with powerful representation capabilities. Spikformer (Zhou et al., 2023) pioneered this integration with Spiking Self Attention (SSA), achieving the first spike-driven computation in Transformers. Spike-driven Transformer (Yao et al., 2023) advanced this concept with a linear-complexity self-attention mechanism that significantly reduces energy requirements. This architecture was later expanded into Spike-driven Transformer V2 (Yao et al., 2024) to enhance versatility across vision tasks. Spike-driven Transformer V3 (Yao et al., 2025) addressed inherent limitations in spiking neurons through Spike Firing Approximation, enabling integer training and spike-driven inference while improving accuracy and efficiency. Beyond these models, numerous excellent Spiking Transformers exist in the vision domain, including QKformer (Zhou et al., 2024), SNN-ViT (Wang et al., 2025), and $\alpha$-SSA-ViT (Xiao et al., 2025). For language modeling, SpikeLM (Xing et al., 2024b) presented the first fully spiking mechanism for general language tasks with bi-directional and elastic spike encoding, narrowing the performance gap between SNNs and ANNs. Despite sophisticated model designs in these works, positional encoding remains largely unexplored. Convolutional layers prevalent in vision Spiking Transformers can serve as implicit positional encoding due to their inherent positional bias. However, this approach is unsuitable for sequence tasks. Sequence-oriented models like SpikeLM also lack specialized positional encoding designs. Therefore, developing dedicated positional encoding techniques for Spiking Transformers is essential.

### 2.2. Positional Encoding

In sequence tasks, sequential information is crucial, yet Transformers exhibit limited capability in capturing sequential information, necessitating the incorporation of positional encoding to address this deficiency. Positional encoding primarily comprises two categories: absolute positional encoding and relative positional encoding. Absolute positional encoding directly integrates positional information into the input. The original absolute positional encoding was generated through a predefined function (Vaswani et al., 2017), subsequently followed by learnable absolute positional encoding (Devlin et al., 2019; Lan et al., 2020). Since absolute positional encoding cannot capture relative posi-

tional information between different tokens, relative positional encoding was proposed. Relative positional encoding methods typically encode the relative position information into the attention mechanism. For instance, in (Shaw et al., 2018), trainable parameters representing relative positions are incorporated when computing attention scores and weighted values. RoPE (Su et al., 2024) combines absolute and relative positional encoding, offering both computational efficiency and relative positional awareness, and has become widely adopted in LLMs. The aforementioned positional encoding methods from the ANN domain cannot be directly applied to SNNs, as they would compromise the spike-driven characteristics of SNNs. Existing positional encoding methods in the SNN domain, such as CPG-PE (Lv et al., 2024) and Gray-PE, Log-PE (Lv et al., 2025b), either cannot represent relative positional information or are incompatible with the linear attention mechanism of Spiking Transformers. Therefore, it is essential to develop a positional encoding method for Spiking Transformers that can represent relative positional information while being compatible with their linear attention mechanism.

# 3. Preliminaries

## 3.1. Leaky Integrate-and-Fire Neuron

SNNs rely on spiking neurons as their processing units. These models aim to replicate the information processing capabilities of biological neurons. Several prominent examples include the Hodgkin-Huxley (Hodgkin & Huxley, 1952), Izhikevich (Izhikevich, 2003), Leaky Integrate-and-Fire (LIF) neurons (Wu et al., 2018), and others (Zhang et al., 2021; Wei et al., 2023). Due to its computational efficiency, the LIF model is widely adopted in Spiking Transformers and has become the most mainstream neuron model. Its membrane potential, a key element of a neuron's firing behavior, is mathematically described as follows:

$$\hat{u}_{i,j}(t+1) = \tau u_{i,j}(t) + I_{i,j}(t), \tag{1}$$

where $u_{i,j}(t)$ and $I_{i,j}(t)$ represent the membrane potential and pre-synaptic input at time $t$ of the neuron in the $i$-th row and $j$-th column of the neuron layer, respectively, for $i = 1, 2, ..., N$ and $j = 1, 2, ..., D$. $\hat{u}_{i,j}(t)$ is the intermediate representation of $u_{i,j}(t)$, and $\tau$ is the constant leaky factor. The size of the neuron layer is $N \times D$, where $N$ represents the number of tokens and $D$ represents the dimension of vectors. When the membrane potential exceeds the threshold, the neuron fires a spike. Therefore, given the threshold $\theta_{i,j}$ of the neuron in the $i$-th row and $j$-th column, the firing function can be described as,

$$s_{i,j}(t+1) = \Theta\left(\hat{u}_{i,j}(t+1) - \theta_{i,j}\right), \tag{2}$$

where $\Theta$ represents the Heaviside step function and $s_{i,j}(t)$ represents the output spike. After spike firing, the membrane

potential will be reset. Common reset methods are divided into two types: hard reset and soft reset, which can be described by Equation 3 and Equation 4, respectively.

$$u_{i,j}(t+1) = [1 - s_{i,j}(t+1)]\,\hat{u}_{i,j}(t+1), \tag{3}$$

$$u_{i,j}(t+1) = \hat{u}_{i,j}(t+1) - s_{i,j}(t+1) \cdot \theta_{i,j}, \tag{4}$$

## 3.2. Spiking Self-Attention

In Spikformer, a spiking version of self-attention named SSA is proposed, which is more suitable for SNNs than vanilla self-attention. Given an input feature sequence $\mathbf{X} \in \mathbb{R}^{T \times N \times D}$, the SSA has three key components, namely query ($\mathbf{Q}$), key ($\mathbf{K}$), and value ($\mathbf{V}$) which are calculated by learnable linear matrices $W_\mathbf{Q}, W_\mathbf{K}, W_\mathbf{V} \in \mathbb{R}^{D \times D}$ and $\mathbf{X}$:

$$\mathbf{Q} = \mathrm{SN}(\mathrm{BN}(\mathrm{Linear_Q}(\mathbf{X}))), \quad \mathbf{Q} \in \mathbb{R}^{T \times N \times D}, \tag{5}$$

$$\mathbf{K} = \mathrm{SN}(\mathrm{BN}(\mathrm{Linear_K}(\mathbf{X}))), \quad \mathbf{K} \in \mathbb{R}^{T \times N \times D}, \tag{6}$$

$$\mathbf{V} = \mathrm{SN}(\mathrm{BN}(\mathrm{Linear_V}(\mathbf{X}))), \quad \mathbf{V} \in \mathbb{R}^{T \times N \times D}, \tag{7}$$

where SN is a spike neuron layer described in subsection 3.1 and $T$ is the number of time steps. BN represents batch normalization. The output of SSA can be computed as:

$$\mathbf{Score} = \sigma \cdot \mathbf{Q}\mathbf{K}^\top, \quad \mathbf{Attn} = \mathrm{SN}(\mathbf{Score} \cdot \mathbf{V}), \tag{8}$$

$$\mathrm{SSA}(\mathbf{Q}, \mathbf{K}, \mathbf{V}) = \mathrm{SN}(\mathrm{BN}(\mathrm{Linear_{Proj}}(\mathbf{Attn}))), \tag{9}$$

where $\sigma$ is a scaling factor to control the large values of the matrix multiplication results. $\mathbf{Q}\mathbf{K}^\top$ is referred to as the attention map. The formulations described above constitute the original SSA designed for computer vision tasks. Adapting this method to natural language processing tasks requires minor modifications, such as replacing Batch Normalization with Layer Normalization.

A key advantage of SSA lies in its linear attention property, whereby the time complexity of SSA can be approximated as $\mathcal{O}(N)$ when the number of tokens $N$ significantly exceeds the token dimension $D$ (Katharopoulos et al., 2020). This is attributable to the absence of Softmax operations in SSA, which enables the computational order of $\mathbf{Q}\mathbf{K}^\top\mathbf{V}$ to be restructured from computing $\mathbf{Q}\mathbf{K}^\top$ first to computing $\mathbf{K}^\top\mathbf{V}$ first. Consequently, the computational complexity is changed from $\mathcal{O}(N^2D)$ to $\mathcal{O}(ND^2)$. When $N \gg D$, this yields $\mathcal{O}(ND^2) \sim \mathcal{O}(N)$.

# 4. Method

## 4.1. Problem Analysis and Design Principles

Positional information proves crucial for sequential tasks. For instance, the sentences "Dog bites man." and "Man bites dog." convey completely opposite meanings. However, in SSA, changing the order of input tokens affects only the sequence of output tokens, while the token vector values

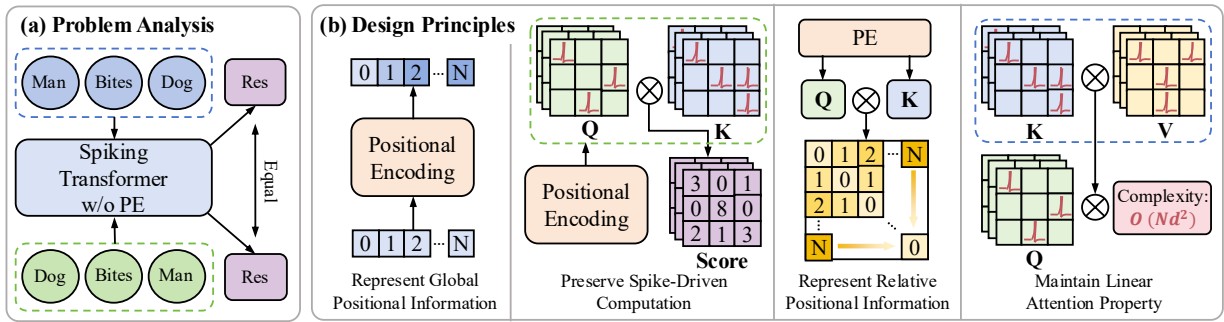

*Figure 1.* Problem analysis and design principles. PE refers to positional encoding.

remain unchanged. Furthermore, the MLP module mixes features only along the channel dimension rather than across tokens, which results in invariant model outputs regardless of token reordering. This behavior is clearly unreasonable for sequence modeling tasks. These observations reveal that the Spiking Transformer model has fundamental limitations in encoding positional information. Therefore, we propose introducing positional encoding mechanisms to overcome the aforementioned shortcomings.

As introduced in subsection 2.2, there have been many works on positional encoding in the ANN field. However, most of these approaches cannot be directly applied to SNN models, as incorporating them into SNNs would compromise the spike-driven characteristics of SNNs. Absolute positional encoding methods, although they can represent global positional information and are simple to implement, cannot represent relative positional information between tokens, which prevents them from capturing the time-translational invariance property in many sequential modeling problems. While relative positional encoding can overcome the aforementioned issues, its implementation relies on attention map computation, causing SSA to lose its linear attention property when relative positional encoding is applied. In summary, we believe that an optimal positional encoding strategy for Spiking Transformers should: (1) represent global positional information to distinguish different positions, (2) preserve the spike-driven characteristics, (3) have the capability to represent relative positional information, and (4) maintain the linear attention property of SSA. These four design principles are illustrated in Figure 1.

### 4.2. Spiking Positional Encoding

We propose Spiking Positional Encoding (SPE), a method that satisfies the four characteristics of an optimal positional encoding strategy for Spiking Transformers outlined in subsection 4.1. The core component of SPE is the Positional Encoding Leaky Integrate-and-Fire (PE-LIF) neuron layer.

The PE-LIF neuron layer builds upon the soft reset LIF neuron layer, with a key modification: each neuron is assigned

a distinct position-dependent threshold $\theta_{i,j}$. Through continuous spike firing and membrane potential reset processes, the positional information encoded in these thresholds is naturally reflected in the emitted spikes. Since neurons activating different tokens possess varying, position-dependent thresholds, the PE-LIF layer inherently encodes global positional information into the activation values.

Specifically, the threshold $\theta_{i,j}$ for the PE-LIF neuron at position $i$ and dimension $j$ is defined as:

$$\theta_{i,j} = \begin{cases} \theta + \lambda \cos\left(i \,/\, 10000^{\frac{j-1}{D}}\right), & \text{j is odd,} \\ \theta + \lambda \sin\left(i \,/\, 10000^{\frac{j-2}{D}}\right), & \text{j is even,} \end{cases} \quad (10)$$

where $i \in \{1, 2, \cdots, N\}$ and $\theta$ is the base threshold, $\lambda$ is a hyperparameter controlling the modulation range around $\theta$, $N$ is the sequence length, and $D$ is the dimension of the PE-LIF layer. Note that $D$ must be even, consistent with standard Spiking Transformer configurations.

As shown in Figure 2, SPE replaces the conventional LIF neuron layers that activate $\mathbf{Q}$ and $\mathbf{K}$ in SSA with PE-LIF neuron layers. We apply PE-LIF to the query and key branches rather than to the value branch for two main reasons. First, relative positional information is primarily required when measuring pairwise token interactions, which are determined by the query-key matching process in self-attention. By encoding position-dependent thresholds into $\mathbf{Q}$ and $\mathbf{K}$, SPE can explicitly modulate token interactions according to their positions, while leaving $\mathbf{V}$ mainly responsible for preserving and carrying semantic content. Second, applying positional encoding directly inside the attention-map computation may break the spike-driven computation paradigm or prevent the associative reordering of $\mathbf{Q}\mathbf{K}^{\top}\mathbf{V}$ into $\mathbf{Q}(\mathbf{K}^{\top}\mathbf{V})$, which is crucial for maintaining linear attention. In contrast, PE-LIF only modifies the spike-generation process of $\mathbf{Q}$ and $\mathbf{K}$, introduces no additional trainable parameters, and preserves the linear attention property of SSA. This design enables SPE to encode relative positional information into spike activations in a computation-friendly manner, as theoretically supported by Proposition 4.1.

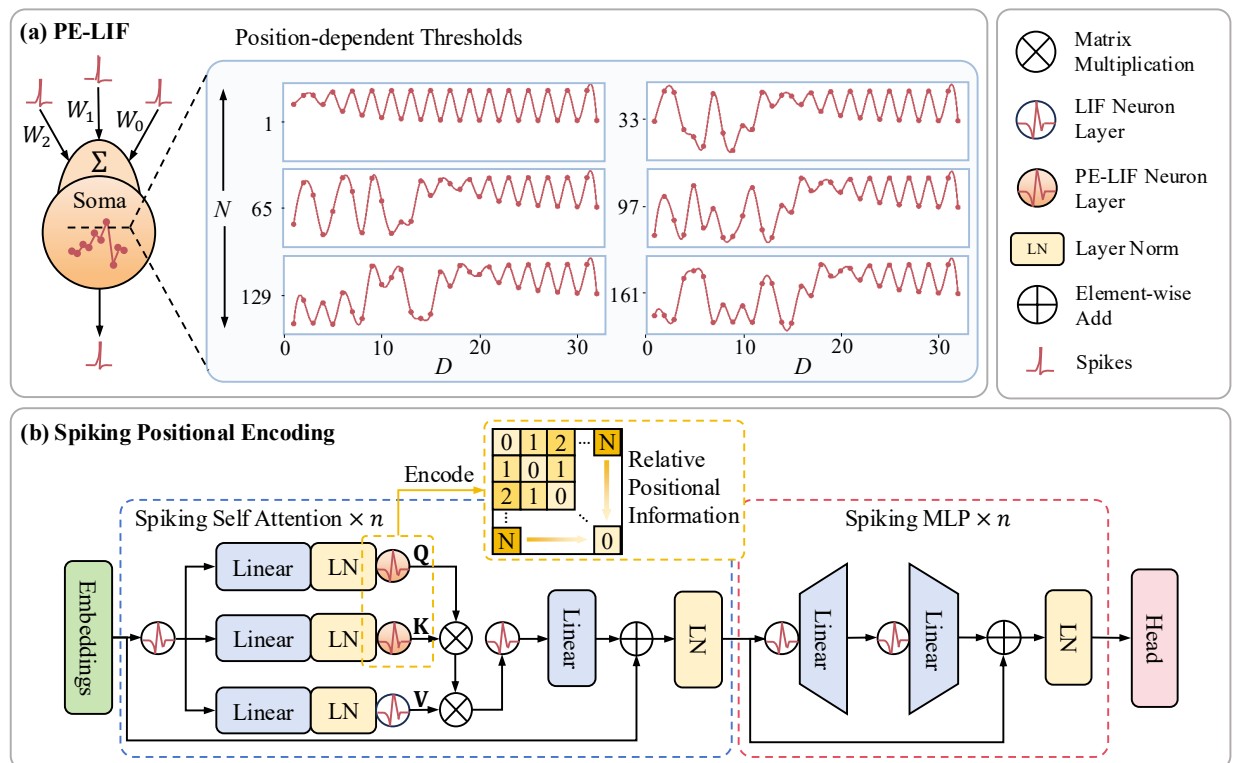

*Figure 2.* Overview of our method. (a) PE-LIF. We visualize the thresholds of PE-LIF neurons at some positions, demonstrating their position-dependent characteristics. (b) Spiking Positional Encoding. We demonstrate how SPE utilizes PE-LIF neuron layers to encode positional information.

**Proposition 4.1.** *(Proof in Appendix A.1) When employing a PE-LIF neuron layer to obtain activations*

$$\begin{cases} \mathbf{Q}(t) = [\boldsymbol{q}_1(t), \boldsymbol{q}_2(t), ..., \boldsymbol{q}_N(t)]^\top \in \{0,1\}^{N \times D}, \\ \mathbf{K}(t) = [\boldsymbol{k}_1(t), \boldsymbol{k}_2(t), ..., \boldsymbol{k}_N(t)]^\top \in \{0,1\}^{N \times D}, \end{cases} \quad (11)$$

*under the condition* $\mathbb{E}\left[\hat{u}_{i,j}(t)\right] = \mathbb{E}\left[s_{i,j}(t)\right]$, *there exists a function* $g$ *that takes* $m - n$ *as its argument such that* $\mathbb{E}\left[\boldsymbol{q}_m^\top(t)\boldsymbol{k}_n(t)\right]$ *contains* $g(m - n)$, *where*

$$g(m - n) =$$
$$\frac{1}{2} \sum_{k=1}^{D/2} \left( \mathbb{E}\left[\mathcal{B}_{2k-1,2k-1}^{(m,n)}\right] + \mathbb{E}\left[\mathcal{B}_{2k,2k}^{(m,n)}\right] \right) \cos(m - n)\beta_k. \quad (12)$$

*Here,* $\mathcal{B}^{(m,n)}$ *and* $\beta_k$ *are defined in Appendix A.1. This demonstrates that PE-LIF can encode relative positional information directly into* $Q(t)$ *and* $K(t)$.

In addition to the four design principles described above, long-term decay is also an essential characteristic that positional encoding capable of representing relative positional information should possess (Su et al., 2024). Intuitively, as the distance between two tokens increases, the degree of association between them should decrease. Therefore,

the value of the function representing the relative positional information between two tokens should diminish as the distance between them increases. This property is referred to as long-term decay. Proposition 4.2 demonstrates that the relative positional information encoded into tokens through PE-LIF possesses this desirable property.

**Proposition 4.2.** *(Proof in Appendix A.2) Under certain conditions, the function* $g(m-n)$ *in Proposition 4.1 satisfies*

$$g(m - n) \sim \frac{D}{4}\text{Re}\left[\int_0^1 \alpha(t)e^{i(m-n)10000^{-t}}dt\right], \quad (13)$$

*where* $\alpha(t)$ *is defined in Appendix A.2.* $i$ *is the imaginary unit, and* $\text{Re}$ *denotes the real part operation. For this oscillatory integral, we have*

$$\lim_{|m-n|\to\infty} \text{Re}\left[\int_0^1 \alpha(t)e^{i(m-n)10000^{-t}}dt\right] = 0, \quad (14)$$

*i.e.,* $g(m - n)$ *exhibits the property of long-term decay.*

### 4.3. Membrane Potential Regularization Surrogate Gradient

The condition $\mathbb{E}\left[\hat{u}_{i,j}(t)\right] = \mathbb{E}\left[s_{i,j}(t)\right]$ serves as a prerequisite for Proposition 4.1 to hold. In general, this condition

*Table 1.* Accuracy (%) comparison between SPE and other positional encoding methods in the SNN domain across six text classification benchmarks. Two datasets are long-text datasets with a maximum sequence length of 1024 during experiments, while the remaining four have a maximum length of 128.

| Model | Param(M) | Max Length = 128 | | | | Avg. | Max Length = 1024 | | Avg. |
|---|---|---|---|---|---|---|---|---|---|
| | | MR | SST-2 | Subj | SST-5 | | AGNEWS | IMDB | |
| BERT | 109.8 | 87.63 | 92.31 | 95.90 | 50.41 | 81.56 | 94.50 | 92.10 | 93.30 |
| Spikformer w/o PE | 109.8 | 75.87 | 81.71 | 91.60 | 41.84 | 72.76 | – | – | – |
| Spikformer w/ CPG-PE | 110.4 | 82.42 | 82.90 | 92.50 | 43.62 | 75.36 | 84.70 | 79.47 | 82.09 |
| Spikformer-XNOR w/o PE | 109.8 | 75.80 | 81.74 | 91.50 | 41.88 | 72.73 | – | – | – |
| Spikformer-XNOR w/ Gray-PE | 109.8 | 83.73 | 84.52 | 92.50 | 44.06 | 76.20 | 84.92 | 79.79 | 82.36 |
| Spikformer-XNOR w/ Log-PE | 109.8 | **83.88** | 84.64 | 92.80 | 44.52 | 76.46 | 86.77 | 80.46 | 83.62 |
| **Spikingformer w/ SPE** | 109.8 | 82.55 | **87.39** | 95.45 | 49.32 | **78.68** | **93.93** | **88.17** | **91.05** |

is not satisfied. To ensure its validity and thereby enable SPE to successfully encode relative positional information, inspired by (Vali et al., 2025), we modify the surrogate gradient of the PE-LIF neuron layers that activate $\mathbf{Q}$ and $\mathbf{K}$ whose forward pass is defined as:

$$\begin{aligned} \boldsymbol{s}' =&\sigma(\hat{\boldsymbol{u}}(t) - \boldsymbol{\theta}) + \|\mathbb{E}_{\mathcal{B}}\left[\boldsymbol{s}(t)\right] - \mathbb{E}_{\mathcal{B}}\left[\hat{\boldsymbol{u}}(t)\right]\|_2 \\ &\cdot \text{sg}\left(\frac{\boldsymbol{s} - \sigma(\hat{\boldsymbol{u}}(t) - \boldsymbol{\theta})}{\|\mathbb{E}_{\mathcal{B}}\left[\boldsymbol{s}(t)\right] - \mathbb{E}_{\mathcal{B}}\left[\hat{\boldsymbol{u}}(t)\right]\|_2}\right) \end{aligned} \quad (15)$$

where $\sigma$ denotes the Sigmoid function, $\mathbb{E}_{\mathcal{B}}\left[\boldsymbol{x}\right] = \frac{1}{\mathcal{B}}\sum_{b=1}^{\mathcal{B}} \boldsymbol{x}_b$ with $\mathcal{B}$ representing the mini-batch size, and $\text{sg}(\cdot)$ is the stop gradient operator that prevents gradients from flowing through the enclosed term during backpropagation. During the forward pass, the output strictly equals the original spike $\boldsymbol{s}$ (i.e., $\boldsymbol{s}' = \boldsymbol{s}$), while during backpropagation, gradients flow through $\hat{\boldsymbol{u}}(t)$ and $\|\mathbb{E}_{\mathcal{B}}\left[\boldsymbol{s}(t)\right] - \mathbb{E}_{\mathcal{B}}\left[\hat{\boldsymbol{u}}(t)\right]\|_2$ but not through the stop-gradient term.

This surrogate gradient promotes the condition $\mathbb{E}\left[\hat{u}_{i,j}(t)\right] = \mathbb{E}\left[s_{i,j}(t)\right]$ to be approximately satisfied, which is supported by Proposition 4.3. We refer to this approach as Membrane Potential Regularization Surrogate Gradient (MPR-SG). Using this method avoids introducing auxiliary loss functions and additional hyperparameters, thereby simplifying the implementation of training code.

**Proposition 4.3.** *(Proof in Appendix A.3) MPR-SG can be interpreted as implicitly introducing an auxiliary loss:*

$$\mathcal{L}_{Aux} \propto \ln \|\mathbb{E}_{\mathcal{B}}\left[\boldsymbol{s}(t)\right] - \mathbb{E}_{\mathcal{B}}\left[\hat{\boldsymbol{u}}(t)\right]\|_2. \quad (16)$$

*MPR-SG encourages the model to minimize the distance between $\mathbb{E}_{\mathcal{B}}\left[\boldsymbol{s}(t)\right]$ and $\mathbb{E}_{\mathcal{B}}\left[\hat{\boldsymbol{u}}(t)\right]$ during training.*

## 5. Experiments

### 5.1. Performance Comparison with State-of-the-Art SNN Positional Encodings

To validate the effectiveness of our proposed method, we compare SPE against several state-of-the-art positional en-

coding methods within the SNN domain, including CPG-PE (Lv et al., 2024), Gray-PE, and Log-PE (Lv et al., 2025b). We evaluated these methods on six widely used text classification benchmarks, categorizing them by sequence length to assess model robustness across varying dependency ranges: MR (Pang & Lee, 2005), SST-2 (Socher et al., 2013), Subj (Pang & Lee, 2004), and SST-5 (Socher et al., 2013) were configured with a maximum sequence length of 128, while AGNEWS (Zhang et al., 2015) and IMDB (Maas et al., 2011) served as long-text classification benchmarks with a maximum sequence length of 1024.

Table 1 presents the comprehensive comparative results. Covering both short- and long-text settings, it provides a clear view of model robustness across sequence lengths. SPE achieves the highest average accuracy among SNN-based methods across both sequence length categories (78.68% for shorter texts and 91.05% for longer texts), out-performing the strongest SNN baseline, Log-PE, by 2.22 points on shorter-text tasks and 7.43 percentage points on longer-text tasks. Notably, in long-text classification, SPE reaches 93.93% on AGNEWS and 88.17% on IMDB, sub-stantially exceeding Log-PE, which achieves 86.77% and 80.46% on these two datasets, respectively. This indicates its strong efficacy in modeling long-range dependencies compared to positional encoding schemes within the SNN domain. On the Subj and AGNEWS datasets, the performance of SPE approaches that of the BERT model, further narrowing the gap between SNNs and ANNs in natural language processing tasks. These results indicate that SPE effectively encodes positional information and enhances the expressive power of Spiking Transformers, particularly in scenarios requiring the modeling of extended sequences.

Moreover, SPE achieves these performance gains without additional parameters, whereas CPG-PE requires 0.6M extra parameters. This efficiency shows that the gains arise from the encoding scheme's enhanced inductive bias rather than increased model capacity.

*Table 2.* Comparison of performance on the GLUE development set for the baseline Spiking Transformer with and without SPE, alongside various other SNNs, ANNs and QNNs. BERT$_{3L}$ indicates the 3-layer BERT model.

| Model | Energy $_{(mJ)}$ | MNLI$_{-m/mm}$ | QQP | QNLI | SST-2 | CoLA | STS-B | MRPC | RTE | Avg. |
|---|---|---|---|---|---|---|---|---|---|---|
| BERT$_{base}$ | 51.41 | 83.8/83.4 | 90.5 | 90.7 | 92.3 | 60.0 | 89.4 | 89.8 | 69.3 | 83.2 |
| BERT$_{3L}$ | 12.9 | 77.1/77.1 | 85.2 | 85.8 | 88.1 | 31.7 | 85.7 | 86.4 | 66.4 | 75.9 |
| ELMo | – | 68.6/– | 86.2 | 71.1 | 91.5 | 44.1 | 70.4 | 76.6 | 53.4 | 70.2 |
| BiBERT | – | 66.1/67.5 | 84.8 | 72.6 | 88.7 | 25.4 | 33.6 | 72.5 | 57.4 | 63.2 |
| BiT | – | 77.1/77.5 | 82.9 | 85.7 | 87.7 | 25.1 | 71.1 | 79.7 | 58.8 | 71.0 |
| BiPFT | – | 69.5/70.6 | 83.7 | 81.7 | 86.2 | 22.9 | 80.2 | 76.2 | 66.1 | 70.8 |
| SpikeBERT | 14.30 | 71.4/71.0 | 68.2 | 66.4 | 85.4 | 16.9 | 18.7 | 82.0 | 57.5 | 59.7 |
| PSN-BERT | – | 35.4/35.2 | 0.0 | 50.5 | 50.9 | 0.0 | 6.8 | 81.2 | 52.7 | 34.7 |
| LIF-BERT | 7.98 | 56.8/55.2 | 70.0 | 60.6 | 80.6 | 14.6 | 20.0 | 82.3 | 53.8 | 54.9 |
| Spikingformer | 6.76 | **71.9/72.5** | 84.7 | 76.0 | 87.2 | 24.4 | 54.5 | 79.7 | 55.6 | 66.8 |
| **Spikingformer w/ SPE** | **4.92** | 70.7/72.0 | **85.8** | **82.9** | **87.4** | **33.3** | **79.3** | **83.5** | **58.1** | **72.6** |

## 5.2. Results on GLUE Benchmark

To evaluate the generalization capability of SPE on complex language understanding tasks, we conduct extensive experiments on the GLUE (General Language Understanding Evaluation) benchmark (Wang et al., 2018). GLUE comprises eight diverse natural language understanding tasks: MNLI (Multi-Genre Natural Language Inference), QQP (Quora Question Pairs), QNLI (Question Natural Language Inference), SST-2 (Stanford Sentiment Treebank), CoLA (Corpus of Linguistic Acceptability), STS-B (Semantic Textual Similarity Benchmark), MRPC (Microsoft Research Paraphrase Corpus), and RTE (Recognizing Textual Entailment). These tasks cover sentence classification, semantic similarity, paraphrase detection, and natural language inference, providing a comprehensive evaluation setting. Experimental settings are provided in Appendix B.

We benchmark our method against a diverse set of models, which are categorized into three groups: (1) Standard ANNs, including BERT$_{base}$, BERT$_{3L}$ (Devlin et al., 2019), and ELMo (Peters et al., 2018); (2) Quantized Neural Networks (QNNs), such as BiBERT (Qin et al., 2022), BiT (Liu et al., 2022), and BiPFT (Xing et al., 2024a); and (3) SNN-based language models, including SpikeBERT (Lv et al., 2025a), PSN-BERT, LIF-BERT (Xing et al., 2024b), and the baseline Spikingformer (Zhou et al., 2026) without SPE.

The results are summarized in Table 2. Spikingformer with SPE achieves an average score of 72.6, surpassing the baseline Spikingformer (66.8) by a significant margin of 5.8 points. The improvement is particularly pronounced in tasks requiring fine-grained semantic or structural modeling. Specifically, SPE improves the baseline by 8.9 points on CoLA and 24.8 points on STS-B, validating our theoretical analysis that SPE captures relative positional information. Furthermore, our method outperforms all competing SNN

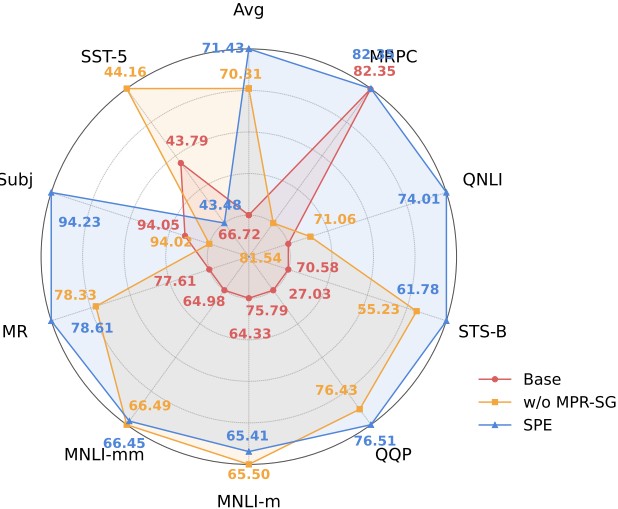

*Figure 3.* Ablation results for individual components of SPE on selected downstream benchmarks.

and QNN baselines and exceeds the performance of the BERT$_{3L}$ model on several tasks. Notably, these gains are realized without introducing additional trainable parameters, underscoring the architectural efficiency of SPE.

In terms of energy efficiency, our model consumes only 4.92 mJ, marking a 27.2% reduction over the baseline (6.76 mJ). This improvement originates from the threshold configuration of PE-LIF neurons (mean > 1) used for computing queries and keys. These higher thresholds inhibit neuronal firing, yielding lower average firing rates for Q and K activations relative to the baseline using unit-threshold LIF neurons. This reduction propagates through the attention mechanism, decreasing mean attention map values and consequently lowering the overall network firing rate.

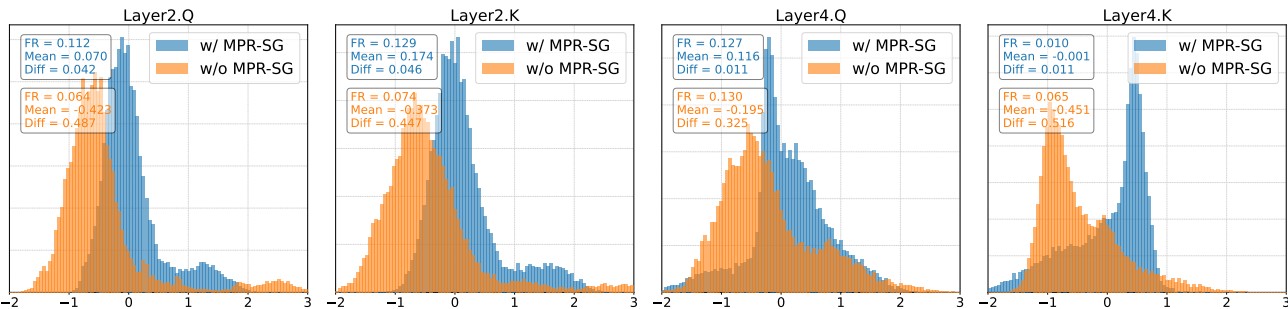

*Figure 4.* Comparison of membrane potential distributions in PE-LIF neurons with and without MPR-SG. FR denotes firing rate. Mean refers to the average membrane potential. Diff represents the absolute difference between FR and Mean.

## 5.3. Ablation Study

To investigate the contribution of each component in SPE, we conduct ablation experiments on eight downstream NLP benchmarks, consistently employing a 4-layer Spiking-former with a hidden dimension of 768. This controlled setting allows us to isolate the effects of PE-LIF and MPR-SG under the same backbone architecture. To reduce the influence of randomness, all experiments are repeated with three random seeds, and we report the average accuracy across these runs. We compare three variants: the baseline Spikingformer without positional encoding, SPE without MPR-SG (w/o MPR-SG), and the full SPE model.

As shown in Fig. 3, introducing the PE-LIF-based positional encoding consistently improves the overall performance over the baseline. The average score increases from 66.72 to 70.31 when MPR-SG is removed, indicating that the position-dependent thresholds in PE-LIF already provide an effective positional inductive bias for Spiking Transformers. The improvement is particularly significant on STS-B, where the score increases from 27.03 to 55.23, suggesting that explicit positional information is crucial for tasks requiring fine-grained semantic matching. Similar gains are also observed on QQP, MNLI-m/mm, MR, and SST-5, further demonstrating the general effectiveness and robustness of the proposed positional encoding mechanism across different task types.

The full SPE model further improves the average score from 70.31 to 71.43. Compared with the variant without MPR-SG, SPE achieves clear gains on QNLI and STS-B, improving the scores from 71.06 to 74.01 and from 55.23 to 61.78, respectively. These results verify the effectiveness of MPR-SG, which encourages the expected membrane potential to better match the expected spike output and thereby facilitates the encoding of relative positional information in PE-LIF neurons. Although minor fluctuations are observed on a few datasets, such as SST-5, the full SPE model achieves the best overall performance and outperforms the baseline on most benchmarks, confirming the complementary contributions of PE-LIF and MPR-SG.

## 5.4. Impact of MPR-SG

Figure 4 illustrates the membrane potential distributions of the PE-LIF neurons responsible for activating $\mathbf{Q}$ and $\mathbf{K}$ during the ablation study. The metric 'Diff' denotes the magnitude of $|\mathbb{E}_{\mathcal{B}}[\hat{u}_{i,j}(t)] - \mathbb{E}_{\mathcal{B}}[s_{i,j}(t)]|$. As observed, the application of MPR-SG shifts the expected membrane potential, $\mathbb{E}[\hat{u}_{i,j}(t)]$, closer to $\mathbb{E}[s_{i,j}(t)]$ compared to the SPE variant without MPR-SG. This empirical observation supports the intended regularization effect of MPR-SG, although it does not constitute a point-wise verification of Proposition 4.3. Additional experimental results are provided in Appendix D.

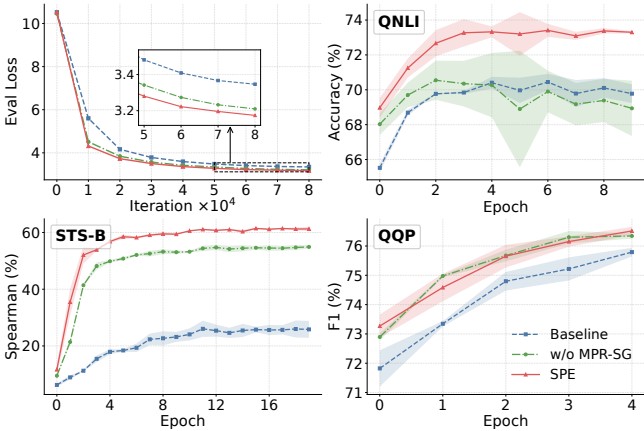

*Figure 5.* Comparison of convergence rate for the Spikingformer baseline, SPE without MPR-SG and SPE.

## 5.5. Rapid Convergence Characteristics of SPE

We recorded the evolution of evaluation loss for the Spikingformer baseline, SPE without MPR-SG, and the full SPE-enhanced model during the pre-training phase of our ablation experiments. In addition, we report the fine-tuning performance curves on QNLI, STS-B, and QQP to further examine the convergence behavior of different methods on downstream tasks, as illustrated in Fig. 5. Notably, the model utilizing SPE exhibits a significantly faster conver-

gence rate and consistently achieves better performance than the baseline and the variant without MPR-SG across both pre-training and fine-tuning settings. These results suggest that SPE improves early-stage optimization and leads to faster empirical convergence. We attribute this improvement to the positional inductive bias introduced by PE-LIF and the stabilizing effect of MPR-SG, while a more fine-grained causal decomposition is left for future investigation.

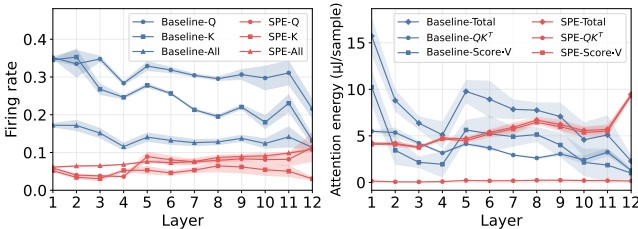

*Figure 6.* Energy reduction analysis of SPE.

### 5.6. Energy Reduction Analysis

The left plot in Fig. 6 shows the layer-wise firing-rate profiles of the baseline and SPE. SPE consistently yields much lower firing rates in the $\mathbf{Q}$ and $\mathbf{K}$ branches, while the gap in the overall layer-wise firing rate is more moderate. This indicates that the spike reduction is mainly concentrated on the $\mathbf{Q}/\mathbf{K}$ branches rather than uniformly distributed across the whole network. Since SPE replaces the LIF neurons of $\mathbf{Q}$ and $\mathbf{K}$ with PE-LIF neurons whose position-dependent thresholds have a larger average value, these results verify that SPE directly suppresses redundant $\mathbf{Q}/\mathbf{K}$ spikes.

The right plot in Fig. 6 decomposes the attention energy into the $\mathbf{QK}^\top$ and $\mathbf{Score} \cdot \mathbf{V}$ terms. While the two terms contribute comparably in the baseline, SPE substantially reduces the $\mathbf{QK}^\top$ energy and lowers the total attention energy from 0.0894 to 0.0666 mJ/sample. Since lower firing rates in the $\mathbf{Q}/\mathbf{K}$ branches generally reduce the effective AC operations for computing $\mathbf{QK}^\top$, this result suggests that SPE improves attention energy efficiency mainly by suppressing redundant $\mathbf{Q}/\mathbf{K}$ spikes.

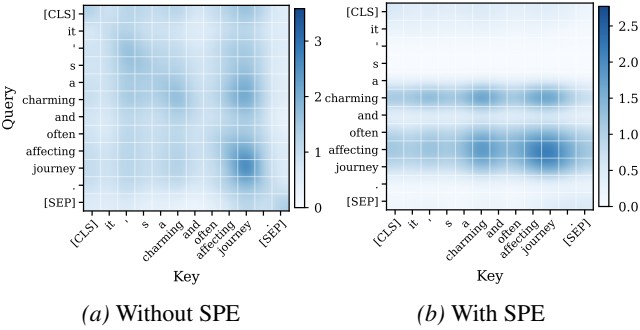

*(a)* Without SPE      *(b)* With SPE

*Figure 7.* Comparison of token selectivity in the attention mechanism for the Spikingformer with and without SPE.

### 5.7. Analysis of Token Selectivity

To analyze the token selectivity of SPE, we visualize the $\mathbf{QK}^\top$ maps of the 4-layer Spikingformer baseline and SPE in the ablation study, as shown in Fig. 7. Compared with the baseline, where attention activity remains broadly distributed over later key positions, SPE suppresses the query activity of less informative tokens and preserves strong responses mainly around sentiment-bearing words such as "charming", "affecting", and "journey". This suggests that the position-dependent PE-LIF neurons act as a token-position gate, reducing redundant attention activity while retaining task-relevant token interactions.

## 6. Conclusion

In this work, we identify the critical need for positional encoding in Spiking Transformers and establish four essential design principles tailored to the spike-driven paradigm. Guided by these principles, we propose SPE, which leverages PE-LIF neuron layers to effectively encode relative positional information through adaptive threshold modulation. We theoretically prove SPE's capability to capture relative positions and its desirable long-term decay properties, demonstrating its full compliance with the established design principles. Comprehensive experiments across thirteen NLP datasets validate that SPE consistently outperforms existing positional encoding methods while maintaining the energy efficiency inherent to SNNs. These results highlight SPE's potential to bridge the performance gap between Spiking Transformers and conventional ANNs, paving the way for more efficient and capable neuromorphic systems for sequential processing tasks. Future work will explore extending SPE to multi-modal Spiking Transformers and hardware-aware implementations on neuromorphic chips.

## Acknowledgements

This work was supported by the National Natural Science Foundation of China (Grants 62576080 and 62220106008), by the Fundamental and Interdisciplinary Disciplines Breakthrough Plan of the Ministry of Education of China (JYB2025XDXM102), the Guangdong Introducing Innovative and Entrepreneurial Teams (Grant 2023ZT10×044), and the Shenzhen Science and Technology Research Fund (Grant JCYJ20220818103001002).

## Impact Statement

This paper presents work whose goal is to advance the field of Machine Learning. There are many potential societal consequences of our work, none of which we feel must be specifically highlighted here.

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

# A. Proof of Propositions

## A.1. Proof of Proposition 4.1

Let

$$\begin{cases} \boldsymbol{q}_m(t) = \left[ s_{m,1}^{(q)}(t), s_{m,2}^{(q)}(t), ..., s_{m,D}^{(q)}(t) \right]^T, \\ \boldsymbol{k}_n(t) = \left[ s_{n,1}^{(k)}(t), s_{n,2}^{(k)}(t), ..., s_{n,D}^{(k)}(t) \right]^T. \end{cases} \tag{17}$$

For analytical tractability, we assume that all $s_{i,j}^{(q)}(t)$ and all $s_{i,j}^{(k)}(t)$ are mutually independent. Then we have

$$\mathbb{E}\left[ \boldsymbol{q}_m^T(t)\boldsymbol{k}_n(t) \right] = \mathbb{E}\left[ \sum_{k=1}^{D} s_{m,k}^{(q)}(t) \cdot s_{n,k}^{(k)}(t) \right] = \sum_{k=1}^{D} \mathbb{E}\left[ s_{m,k}^{(q)}(t) \right] \mathbb{E}\left[ s_{n,k}^{(k)}(t) \right]. \tag{18}$$

Since $\mathbb{E}\left[ \hat{u}_{i,j}(t) \right] = \mathbb{E}[s_{i,j}(t)]$, we have

$$\mathbb{E}\left[ \boldsymbol{q}_m^T(t)\boldsymbol{k}_n(t) \right] = \sum_{k=1}^{D} \mathbb{E}\left[ s_{m,k}^{(q)}(t) \right] \mathbb{E}\left[ s_{n,k}^{(k)}(t) \right] = \sum_{k=1}^{D} \mathbb{E}\left[ \hat{u}_{m,k}^{(q)}(t) \right] \mathbb{E}\left[ \hat{u}_{n,k}^{(k)}(t) \right]. \tag{19}$$

Expanding Equations 1, 2 and 4, we obtain

$$\hat{u}_{i,j}(t) = \sum_{k=0}^{t-1} \tau^{t-1-k} I_{i,j}(k) - \theta_{i,j} \sum_{k=1}^{t-1} \tau^{t-k} s_{i,j}(k), i = 1, 2, ..., N, j = 1, 2, ..., D. \tag{20}$$

Furthermore, we have

$$\begin{aligned} \mathbb{E}\left[ \boldsymbol{q}_m^T(t)\boldsymbol{k}_n(t) \right] = & \mathbb{E}\left[ \boldsymbol{a}_m^{(q)T}(t)\boldsymbol{a}_n^{(k)}(t) \right] - \theta\mathbb{E}\left[ \boldsymbol{a}_m^{(q)T}(t)\boldsymbol{b}_n^{(k)}(t) \right] - \theta\mathbb{E}\left[ \boldsymbol{a}_n^{(k)T}(t)\boldsymbol{b}_m^{(q)}(t) \right] \\ & - \lambda\mathbb{E}\left[ \boldsymbol{a}_m^{(q)T}(t)\left( \boldsymbol{b}_n^{(k)}(t) \odot \boldsymbol{\alpha}_n \right) \right] - \lambda\mathbb{E}\left[ \boldsymbol{a}_n^{(k)T}(t)\left( \boldsymbol{b}_m^{(q)}(t) \odot \boldsymbol{\alpha}_m \right) \right] \\ & + \theta^2\mathbb{E}\left[ \boldsymbol{b}_m^{(q)T}(t)\boldsymbol{b}_n^{(k)}(t) \right] + \theta\lambda\mathbb{E}\left[ \boldsymbol{b}_m^{(q)T}(t)\left( \boldsymbol{b}_n^{(k)}(t) \odot \boldsymbol{\alpha}_n \right) \right] \\ & + \theta\lambda\mathbb{E}\left[ \boldsymbol{b}_n^{(k)T}(t)\left( \boldsymbol{b}_m^{(q)}(t) \odot \boldsymbol{\alpha}_m \right) \right] + \lambda^2\mathbb{E}\left[ \boldsymbol{\alpha}_m^T \mathcal{B}^{(m,n)} \boldsymbol{\alpha}_n \right], \end{aligned} \tag{21}$$

where

$$\begin{cases} \boldsymbol{a}_i^{(\cdot)}(t) = \left[ \sum_{k=0}^{t-1} \tau^{t-1-k} I_{i,1}^{(\cdot)}(k), ..., \sum_{k=0}^{t-1} \tau^{t-1-k} I_{i,D}^{(\cdot)}(k) \right]^T, \\ \boldsymbol{b}_i^{(\cdot)}(t) = \left[ \sum_{k=1}^{t-1} \tau^{t-k} s_{i,1}^{(\cdot)}(k), ..., \sum_{k=1}^{t-1} \tau^{t-k} s_{i,D}^{(\cdot)}(k) \right]^T, \\ \boldsymbol{\alpha}_i = \left[ ..., \cos\left( \frac{i}{10000^{\frac{j-1}{D}}} \right), \sin\left( \frac{i}{10000^{\frac{j-2}{D}}} \right), ... \right]^T, \\ \mathcal{B}^{(i,j)} = \text{diag}\left( \boldsymbol{b}_i^{(q)} \odot \boldsymbol{b}_j^{(k)} \right). \end{cases} \tag{22}$$

Consider the term $\mathbb{E}\left[ \boldsymbol{\alpha}_m^T \mathcal{B}^{(m,n)} \boldsymbol{\alpha}_n \right]$. Applying the product-to-sum formula, we obtain:

$$\begin{aligned} \mathbb{E}\left[ \boldsymbol{\alpha}_m^T \mathcal{B}^{(m,n)} \boldsymbol{\alpha}_n \right] = & \sum_{k=1}^{D/2} \left( \mathbb{E}\left[ \mathcal{B}_{2k-1,2k-1}^{(m,n)} \right] \cos m\beta_k \cos n\beta_k + \mathbb{E}\left[ \mathcal{B}_{2k,2k}^{(m,n)} \right] \sin m\beta_k \sin n\beta_k \right) \\ = & \frac{1}{2} \sum_{k=1}^{D/2} \left( \mathbb{E}\left[ \mathcal{B}_{2k-1,2k-1}^{(m,n)} \right] + \mathbb{E}\left[ \mathcal{B}_{2k,2k}^{(m,n)} \right] \right) \cos(m-n)\beta_k \\ & + \frac{1}{2} \sum_{k=1}^{D/2} \left( \mathbb{E}\left[ \mathcal{B}_{2k-1,2k-1}^{(m,n)} \right] - \mathbb{E}\left[ \mathcal{B}_{2k,2k}^{(m,n)} \right] \right) \cos(m+n)\beta_k, \end{aligned} \tag{23}$$

where $\beta_i = 1/10000^{\frac{2(i-1)}{D}}$. Thus, we find that $g(m-n)$ is:

$$g(m-n) = \frac{1}{2} \sum_{k=1}^{D/2} \left( \mathbb{E}\left[ \mathcal{B}_{2k-1,2k-1}^{(m,n)} \right] + \mathbb{E}\left[ \mathcal{B}_{2k,2k}^{(m,n)} \right] \right) \cos(m-n)\beta_k. \tag{24}$$

Therefore, Proposition 4.1 is proved.

### A.2. Proof of Proposition 4.2

From Proposition 4.1, we have established:

$$g(m-n) = \frac{1}{2} \sum_{k=1}^{D/2} \left( \mathbb{E}\left[ \mathcal{B}_{2k-1,2k-1}^{(m,n)} \right] + \mathbb{E}\left[ \mathcal{B}_{2k,2k}^{(m,n)} \right] \right) \cos(m-n)\beta_k, \tag{25}$$

where $\beta_i = 1/10000^{\frac{2(i-1)}{D}}$. By Euler's formula $e^{ix} = \cos x + i \sin x$, we have:

$$g(m-n) = \frac{1}{2} \sum_{k=1}^{D/2} \left( \mathbb{E}\left[ \mathcal{B}_{2k-1,2k-1}^{(m,n)} \right] + \mathbb{E}\left[ \mathcal{B}_{2k,2k}^{(m,n)} \right] \right) \mathrm{Re}\left[ e^{i(m-n)\beta_k} \right]. \tag{26}$$

Utilizing the linearity of the real part of complex numbers, we obtain:

$$\begin{aligned}
g(m-n) &= \frac{1}{2} \mathrm{Re}\left[ \sum_{k=1}^{D/2} \left( \mathbb{E}\left[ \mathcal{B}_{2k-1,2k-1}^{(m,n)} \right] + \mathbb{E}\left[ \mathcal{B}_{2k,2k}^{(m,n)} \right] \right) e^{i(m-n)\beta_k} \right] \\
&= \frac{D}{4} \mathrm{Re}\left[ \sum_{k=1}^{D/2} \left( \mathbb{E}\left[ \mathcal{B}_{2k-1,2k-1}^{(m,n)} \right] + \mathbb{E}\left[ \mathcal{B}_{2k,2k}^{(m,n)} \right] \right) e^{i(m-n)\beta_k} \frac{2}{D} \right].
\end{aligned} \tag{27}$$

From equation 27, we observe that $g(m-n)$ is precisely a Riemann sum over the interval $[0, 1]$. When $D$ is large and $\alpha$ is sufficiently regular, by the definition of integration, $g(m-n)$ approximates an oscillatory integral, yielding the approximation:

$$g(m-n) \sim \frac{D}{4} \mathrm{Re}\left[ \int_0^1 \alpha(t) e^{i(m-n)10000^{-t}} dt \right], \tag{28}$$

where $t = \frac{2(k-1)}{D}$ and

$$\alpha(t) = \lim_{D \to \infty} \left( \mathbb{E}\left[ \mathcal{B}_{2k-1,2k-1}^{(m,n)} \right] + \mathbb{E}\left[ \mathcal{B}_{2k,2k}^{(m,n)} \right] \right)\Big|_{k = \frac{Dt}{2}+1}. \tag{29}$$

**Lemma A.1.** *(Riemann-Lebesgue). Let $f(x) \in L^1(\mathbb{R})$, meaning $f$ is Lebesgue integrable on $\mathbb{R}$. Define its Fourier transform as*

$$\hat{f}(\xi) = \int_{\mathbb{R}} f(x) e^{-i\xi x} \, dx, \tag{30}$$

*Then $\lim_{|\xi| \to \infty} \hat{f}(\xi) = 0$.*

Consider the integral

$$I(\Delta) = \int_0^1 \alpha(t) e^{-i\Delta 10000^{-t}} dt, \qquad \Delta = n - m. \tag{31}$$

Making the substitution $u = 10000^{-t}$, we have $t = -\log_{10000} u$ and $dt = -\frac{1}{u \ln 10000} du$. As $t$ ranges from 0 to 1, $u$ decreases from 1 to $10000^{-1}$. Thus,

$$I(\Delta) = \int_{u=1}^{10000^{-1}} \alpha\left( -\log_{10000} u \right) e^{-i\Delta u} \left( -\frac{1}{u \ln 10000} \right) du = \int_{10000^{-1}}^1 \beta(u) e^{-i\Delta u} \, du, \tag{32}$$

where we define

$$\beta(u) = \frac{\alpha\left( -\log_{10000} u \right)}{u \ln 10000}. \tag{33}$$

Assuming $\alpha(t)$ is bounded on $[0, 1]$, we have $\beta(u) \in L^1([10000^{-1}, 1])$. Since $1/(u \ln 10000)$ is integrable on $[10000^{-1}, 1]$, $\beta$ is integrable. Therefore, by Lemma A.1, we obtain

$$\lim_{|\Delta| \to \infty} I(\Delta) = 0 \quad \implies \quad \lim_{|\Delta| \to \infty} \mathrm{Re}\, I(\Delta) = 0. \tag{34}$$

Therefore, Proposition 4.2 is proved.

## A.3. Proof of Proposition 4.3

To simplify notation, we omit the time step $t$ and batch indices. We rewrite the definition of $s'$ as:

$$s' = \sigma(\hat{u} - \theta) + r \cdot \text{sg}\left(\frac{s - \sigma(\hat{u} - \theta)}{r}\right) \tag{35}$$

where $r = \|\mathbb{E}_{\mathcal{B}}[s] - \mathbb{E}_{\mathcal{B}}[\hat{u}]\|_2$ denotes the $\ell_2$ distance between the batch-averaged spike output and membrane potential.

Consider the total loss function $\mathcal{L}$. Its differential with respect to $s'$ is:

$$d\mathcal{L} = \langle \nabla_{s'}\mathcal{L}, ds' \rangle \tag{36}$$

During backpropagation, the stop-gradient operator $\text{sg}(\cdot)$ prevents gradients from flowing through the enclosed term. Consequently, the differential of $s'$ with respect to model parameters is:

$$ds' = d\sigma(\hat{u} - \theta) + dr \cdot \frac{s - \sigma(\hat{u} - \theta)}{r} \tag{37}$$

Substituting this into the loss differential yields:

$$
\begin{aligned}
d\mathcal{L} &= \left\langle \nabla_{s'}\mathcal{L}, d\sigma(\hat{u} - \theta) + dr \cdot \frac{s - \sigma(\hat{u} - \theta)}{r} \right\rangle \\
&= \langle \nabla_{s'}\mathcal{L}, d\sigma(\hat{u} - \theta) \rangle + \langle \nabla_{s'}\mathcal{L}, s - \sigma(\hat{u} - \theta) \rangle \frac{dr}{r}
\end{aligned}
\tag{38}
$$

Observing that $\frac{dr}{r} = d(\ln r)$, we obtain:

$$d\mathcal{L} = \langle \nabla_{s'}\mathcal{L}, d\sigma(\hat{u} - \theta) \rangle + \langle \nabla_{s'}\mathcal{L}, s - \sigma(\hat{u} - \theta) \rangle \, d(\ln r) \tag{39}$$

The first term, $\langle \nabla_{s'}\mathcal{L}, d\sigma(\hat{u}-\theta) \rangle$, represents the standard gradient flow identical to conventional surrogate gradient methods. The second term, $\langle \nabla_{s'}\mathcal{L}, s - \sigma(\hat{u} - \theta) \rangle \, d(\ln r)$, constitutes the additional gradient contribution introduced by MPR-SG. This term is equivalent to implicitly augmenting the original loss with an auxiliary loss:

$$\mathcal{L}_{Aux} = \text{sg}\left[\langle \nabla_{s'}\mathcal{L}, s - \sigma(\hat{u} - \theta) \rangle\right] \cdot \ln r \tag{40}$$

where $\text{sg}[\cdot]$ indicates that the coefficient does not participate in higher-order gradient computations.

Therefore, we have:

$$\mathcal{L}_{Aux} \propto \ln \|\mathbb{E}_{\mathcal{B}}[s(t)] - \mathbb{E}_{\mathcal{B}}[\hat{u}(t)]\|_2 \tag{41}$$

This auxiliary loss incentivizes the model to minimize $\|\mathbb{E}_{\mathcal{B}}[s(t)] - \mathbb{E}_{\mathcal{B}}[\hat{u}(t)]\|_2$, thereby driving $\mathbb{E}[s(t)]$ and $\mathbb{E}[\hat{u}(t)]$ progressively closer during training.

*Remark* A.2. The effectiveness of this approach relies on the coefficient $\langle \nabla_{s'}\mathcal{L}, s - \sigma(\hat{u} - \theta) \rangle$ being positive during training. To see why this condition typically holds, we consider a first-order Taylor approximation of the loss function $\mathcal{L}$ at $s'$ in the neighborhood of $\sigma(\hat{u} - \theta)$:

$$\mathcal{L}(s) \approx \mathcal{L}(\sigma(\hat{u} - \theta)) + \langle \nabla_{s'}\mathcal{L}, s - \sigma(\hat{u} - \theta) \rangle \tag{42}$$

Rearranging gives:

$$\langle \nabla_{s'}\mathcal{L}, s - \sigma(\hat{u} - \theta) \rangle \approx \mathcal{L}(s) - \mathcal{L}(\sigma(\hat{u} - \theta)) \tag{43}$$

Since $\sigma(\hat{u} - \theta)$ provides a continuous-valued approximation that retains richer gradient information compared to the discrete, binary spikes $s$, it serves as a more informative representation for optimization. Specifically, the binarization process inherent in spike generation (converting continuous membrane potentials to discrete 0/1 values) constitutes an information-lossy operation. Consequently, using the discrete spike representation $s$ directly typically results in higher task loss than using the continuous approximation $\sigma(\hat{u} - \theta)$, i.e., $\mathcal{L}(s) > \mathcal{L}(\sigma(\hat{u} - \theta))$.

This inequality implies $\langle \nabla_{s'}\mathcal{L}, s - \sigma(\hat{u} - \theta) \rangle > 0$ with high probability, ensuring that the auxiliary loss term correctly encourages reduction of $r = \|\mathbb{E}_{\mathcal{B}}[s] - \mathbb{E}_{\mathcal{B}}[\hat{u}]\|_2$, thereby facilitating the satisfaction of the condition $\mathbb{E}[\hat{u}_{i,j}(t)] = \mathbb{E}[s_{i,j}(t)]$ required by Proposition 4.1.

# B. Datasets and Experimental Settings

## B.1. Datasets

**Movie Reviews (MR)**  The MR dataset, originally compiled by Pang and Lee from Rotten Tomatoes, comprises binary sentiment classification data drawn from movie reviews. It contains 10,662 sentences, evenly split between positive and negative sentiments, serving as a standard benchmark for evaluating sentiment polarity detection in natural language processing systems. The dataset presents challenges due to the subtle and context-dependent nature of sentiment expressed in film criticism.

**Subjectivity Dataset (Subj)**  The Subj dataset is designed for binary subjectivity detection, distinguishing between subjective expressions (opinions, emotions, speculations) and objective statements (factual descriptions). Extracted from Rotten Tomatoes and plot summaries from the Internet Movie Database, the dataset contains 10,000 sentences balanced across both classes. This task evaluates a model's capacity to identify the presence of personal perspective versus factual content in text.

**Stanford Sentiment Treebank (SST)**  The SST comprises sentence-level movie reviews from Rotten Tomatoes that have been parsed with the Penn Treebank syntactic structure. The corpus supports two granularity levels: SST-2, which provides binary labels (positive/negative) for each of its 67,000+ sentences and evaluates models via classification accuracy; and SST-5, the fine-grained variant that scores every constituent in the parse tree on a five-class ordinal scale (very negative, negative, neutral, positive, very positive) across 11,855 sentences. Together, these tasks benchmark both coarse sentiment polarity detection and the modelling of compositional semantics and intensity nuances.

**AG's News Corpus (AGNEWS)**  AGNEWS comprises categorized news articles collected from the AG news corpus, widely utilized for topic classification benchmarking. The dataset is organized into four balanced categories: World, Sports, Business, and Science/Technology, with 30,000 training samples and 1,900 test samples per class. News articles present distinct challenges including diverse vocabulary, named entities, and varied document lengths, making this corpus valuable for assessing general text categorization capabilities.

**IMDB**  The IMDB dataset is a large-scale binary sentiment classification benchmark derived from the Internet Movie Database, containing 50,000 highly polar movie reviews split equally between training and test sets. With relatively long reviews, this dataset tests a model's ability to capture long-range dependencies and contextual sentiment shifts across lengthy documents, providing a more challenging evaluation scenario than shorter text benchmarks.

**The Corpus of Linguistic Acceptability (CoLA)**  CoLA (Warstadt et al., 2019) comprises approximately 10,000 English sentences meticulously collected from linguistic literature, annotated for grammatical acceptability by expert linguists. Given a single sentence as input, the task requires binary classification to determine whether the sentence adheres to principles of Standard English grammar. Performance is evaluated using the Matthews Correlation Coefficient (MCC), which accounts for the class imbalance inherent in grammatical acceptability judgments.

**The Microsoft Research Paraphrase Corpus (MRPC)**  MRPC (Dolan & Brockett, 2005) consists of 5,800 sentence pairs extracted from online news sources, professionally annotated to identify semantic equivalence (paraphrases) versus non-equivalent pairs. This binary classification task evaluates a model's capacity to recognize semantic similarity regardless of lexical and syntactic variations. Both F1 score and accuracy are reported to account for class distribution imbalance.

**The Semantic Textual Similarity Benchmark (STS-B)**  Comprised of sentence pairs drawn from news headlines, video captions, and image descriptions, STS-B (Cer et al., 2017) requires predicting the degree of semantic equivalence between two texts as a continuous score ranging from 0 to 5. Annotated human judgments provide gold-standard similarity ratings. Model performance is assessed using the Pearson correlation coefficient between predicted and ground-truth scores.

**The Quora Question Pairs (QQP)**  QQP comprises 600,000 question pairs sampled from the Quora website, annotated to determine whether two questions are semantically duplicate despite varying surface forms. As a binary classification task, it tests semantic understanding and paraphrase detection capabilities in the domain of social question-answering. Evaluation employs both F1 score and accuracy metrics.

**The Multi-Genre Natural Language Inference (MNLI)**  As one of the largest NLI datasets, MNLI (Williams et al., 2018) contains 433,000 sentence pairs spanning diverse genres including fiction, government documents, and telephone conversations. Each pair consists of a premise and hypothesis to be classified into one of three entailment relations: entailment, contradiction, or neutral. The evaluation protocol includes both a matched test set (same genres as training) and a mismatched test set (cross-genre generalization), with accuracy as the primary metric.

**The Question-answering NLI (QNLI)**  Converted from the Stanford Question Answering Dataset (Rajpurkar et al., 2016), QNLI reformulates reading comprehension as textual entailment by constructing sentence pairs from questions and corresponding context sentences. The binary classification task requires determining whether a context sentence contains the answer to a given question, thereby evaluating models' ability to perform fine-grained question-passage entailment recognition.

**The Recognizing Textual Entailment (RTE)**  Compiled from a series of annual RTE challenges (Dagan et al., 2005; Haim et al., 2006; Giampiccolo et al., 2007; Bentivogli et al., 2009), this dataset aggregates text entailment pairs from various sources including newswire and Wikipedia. Given a premise-hypothesis pair, the task requires binary classification to identify whether the hypothesis can be logically inferred from the premise. Despite its smaller scale compared to MNLI, RTE remains a challenging benchmark requiring robust semantic reasoning capabilities.

### B.2. Experimental Settings

Following the standard BERT paradigm, we implement Spikingformer with SPE via a two-stage approach: pre-training and fine-tuning.

**Pre-training Configuration**  We conduct masked language modeling (MLM) pre-training with a masking probability of 15%. The global batch size is set to 512, distributed across 8 nodes with 64 sequences per device and a gradient accumulation step of 1. We employ the AdamW optimizer with an initial learning rate of $2 \times 10^{-4}$ and without weight decay. The learning rate schedule follows a linear decay after a linear warm-up period of 5,000 steps. Each sequence contains a maximum of 128 tokens, resulting in approximately 42.6 billion training tokens processed in total. All datasets are accessed via the Hugging Face Datasets library: Stories[1], BookCorpus[2], CC-News[3], OpenWebText[4], and Wikipedia[5].

**Fine-tuning Configuration**  We fine-tune the pre-trained model on individual tasks using the provided pre-trained checkpoint as initialization. We train all models with a per-device batch size of 32. The maximum sequence length is set to 1024 tokens for AGNEWS and IMDB, and 128 tokens for the other tasks. We employ the AdamW optimizer with a learning rate of $2 \times 10^{-5}$ and without weight decay. A linear learning rate decay schedule is utilized without warmup steps, and gradient accumulation is disabled.

## C. Raw Results of the Ablation Study

To provide a more detailed view of the ablation study, we report the raw results obtained with three random seeds in Table 3. All variants are evaluated under the same experimental settings as those used in Section 5.3. The "Base" model denotes the Spikingformer baseline without positional encoding, "w/o MPR-SG" denotes SPE without the proposed membrane potential regularization surrogate gradient, and "SPE" denotes the full model. Compared with the baseline, introducing PE-LIF generally improves performance across different seeds, with particularly large gains on STS-B, QNLI, QQP, and MNLI. The full SPE model further improves over the variant without MPR-SG on QNLI and STS-B for all three seeds, indicating that MPR-SG helps PE-LIF encode positional information more effectively. Although small fluctuations are observed on a few datasets such as MRPC and SST-5, the overall trend remains consistent with the averaged ablation results reported in the main text.

---

[1]https://huggingface.co/datasets/roneneldan/TinyStories
[2]https://huggingface.co/datasets/bookcorpus
[3]https://huggingface.co/datasets/vblagoje/cc_news
[4]https://huggingface.co/datasets/Skylion007/openwebtext
[5]https://huggingface.co/datasets/wikimedia/wikipedia

*Table 3.* Raw ablation results on selected downstream benchmarks across three random seeds.

| Seed | Method | MRPC | QNLI | STS-B | QQP | MNLI-m/mm | MR | Subj | SST-5 |
|------|--------|------|------|-------|-----|-----------|-----|------|-------|
| 41 | Base | 82.86 | 70.77 | 24.55 | 75.72 | 64.35/65.07 | 77.86 | 94.45 | 43.71 |
| | w/o MPR-SG | 81.38 | 72.16 | 55.26 | 76.27 | 65.22/66.79 | 77.86 | 94.10 | 44.12 |
| | SPE | 82.59 | 73.99 | 61.51 | 76.53 | 65.28/66.47 | 77.86 | 94.10 | 43.80 |
| 42 | Base | 82.08 | 70.09 | 30.07 | 75.99 | 63.97/64.81 | 77.58 | 93.65 | 44.30 |
| | w/o MPR-SG | 81.79 | 69.21 | 55.98 | 76.47 | 65.75/66.05 | 78.99 | 94.25 | 44.34 |
| | SPE | 82.30 | 73.64 | 62.27 | 76.68 | 65.29/66.45 | 78.52 | 94.25 | 43.26 |
| 43 | Base | 82.11 | 70.88 | 26.47 | 75.66 | 64.67/65.05 | 77.39 | 94.05 | 43.35 |
| | w/o MPR-SG | 81.45 | 71.81 | 54.46 | 76.55 | 65.52/66.63 | 78.14 | 93.70 | 44.03 |
| | SPE | 82.17 | 74.41 | 61.55 | 76.33 | 65.65/66.44 | 79.46 | 94.35 | 43.39 |

# D. More Comparisons of Membrane Potential Distributions

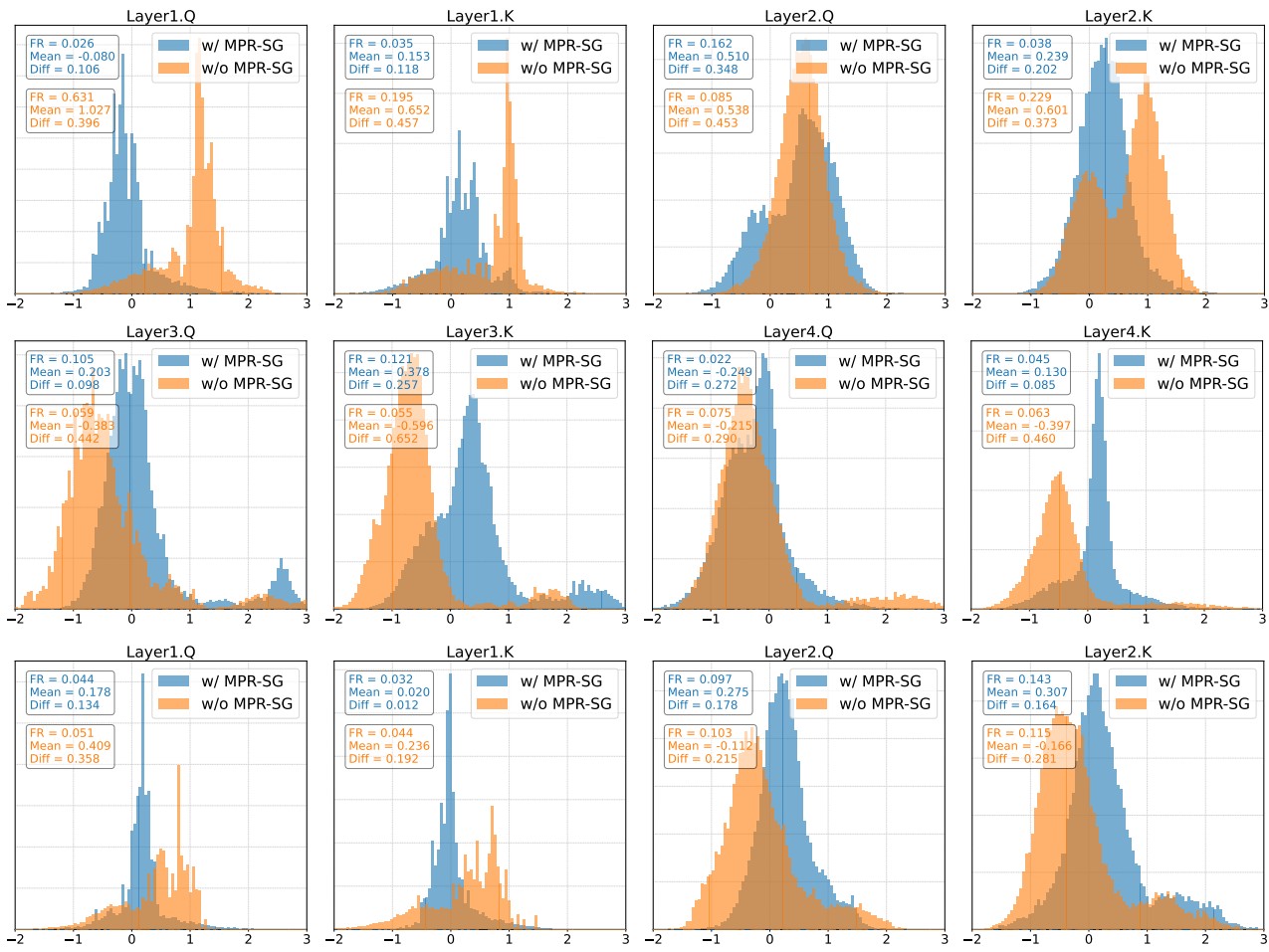

*Figure 8.* More comparisons of membrane potential distributions.

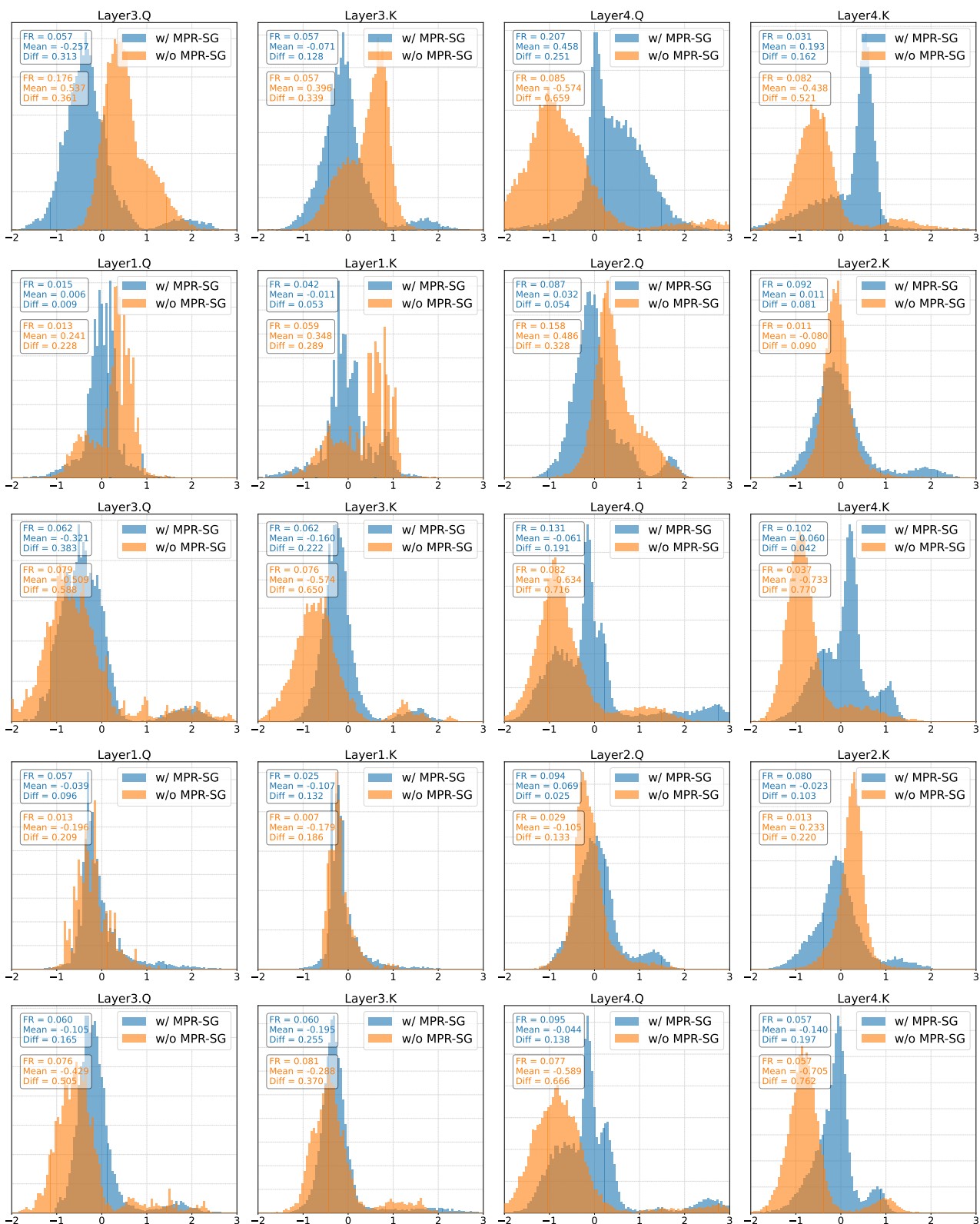

*Figure 8.* More comparisons of membrane potential distributions (continued).

