# OpenReview forum: "Positional Encoding for Spiking Transformers"
_ICML.cc/2026/Conference — ICML 2026 regular_

### Official Review · Reviewer_UhkF · 2026-03-09

**Soundness:** 3
**Presentation:** 2
**Significance:** 3
**Originality:** 3
**Overall Recommendation:** 3
**Confidence:** 2

**Summary:**

This paper addresses a key limitation of Transformer-based SNN models: applying ANN-style Relative Positional Encoding often requires compromising the spike-driven paradigm. To address this issue, the authors propose Spiking Positional Encoding (SPE), designed to capture relative positional information in Spiking Transformers while preserving the spike-driven computational pattern. The paper provides theoretical analysis to support the proposed idea and evaluates the method on a wide range of NLP benchmarks. The experimental results show strong performance and improved energy efficiency.

**Compliance With Llm Reviewing Policy:**

Affirmed.

**Key Questions For Authors:**

1. The paper provides limited details on the experimental setup. Could the authors describe the training and evaluation settings in more detail?
2. In Table 2, how is the energy efficiency of each model measured or calculated? It is currently unclear whether the reported values are from hardware measurements, simulations, or analytical estimates.
3. In Section 5.2 (Results on GLUE Benchmark), the authors explain the energy efficiency improvement by noting that the thresholds of the PE-LIF neurons are, on average, greater than 1. Could the authors clarify whether this benefit mainly comes from the hyperparameter $\lambda$, or whether it is an intrinsic advantage of SPE itself?

**Limitations:**

1. Limited information about the experimental setup reduces the reliability and reproducibility of the results.
2. The ablation study is not sufficient to fully validate the effectiveness of the proposed method.

**Strengths And Weaknesses:**

Strengths
1. The paper provides a theoretical analysis to justify the validity of the proposed idea.
2. The proposed design follows the spike-driven paradigm required by SNN-based architectures in a principled manner.
3. The experiments are conducted on a broad range of benchmarks, including short-text, long-text, and GLUE-style tasks, and show empirical performance improvements across diverse NLP settings.

Weaknesses
1. The ablation study is limited to the presence or absence of MPR-SG, which does not sufficiently demonstrate the effectiveness of each component of the proposed method.
2. The paper provides limited details on hyperparameters and experimental settings, which makes it difficult to assess reproducibility.

---

> ### Author Rebuttal · Authors · 2026-03-31
>
> # Response to Reviewer UhkF
>
> We thank the reviewer for the helpful comments on ablation and reproducibility.
>
> ---
>
> **Weakness 1.** We respectfully note that the current ablation already separates the two key components of SPE: comparing the baseline with "SPE w/o MPR-SG" isolates the effect of PE-LIF itself, while comparing "SPE w/o MPR-SG" with full SPE isolates the additional effect of MPR-SG. The results (64.00 -> 64.68 -> 65.97 average on GLUE dev) show that PE-LIF alone is beneficial, and that MPR-SG is important for realizing the full relative-position benefit. We agree that further ablations, such as varying lambda or applying PE-LIF to only part of the attention block, would strengthen the paper; We will explore these potentials in our future work.
>
> ---
>
> **Weakness 2 / Question 1.** We respectfully note that Appendix B.2 already contains the main training setup. Concretely, we use a two-stage pipeline: MLM pre-training with 15% masking, global batch size 512 over 8 nodes, AdamW, initial learning rate 2e-4, no weight decay, 5k warmup steps, linear decay, maximum length 128, and about 42.6B training tokens from TinyStories, BookCorpus, CC-News, OpenWebText, and Wikipedia; then task-specific fine-tuning with batch size 32 per device, AdamW, learning rate 2e-5, no weight decay, linear decay without warmup, and maximum length 128.
>
> ---
>
> **Question 2.** Table 2 reports analytical energy estimates, not hardware measurements. Specifically, for layer $l$, we compute
> $$SOP_l = fr \times T \times FLOP_l^e$$
> where $fr$ is the firing rate, $T$ is the number of time steps, and $FLOP_l^e$ is the event-driven FLOPs of that layer. The total energy is estimated as
> $$E = E_{AC}\sum_{i=1}^L SOP_i + E_{MAC}\sum_{i=1}^L FLOP_i^{ue}$$
> where $FLOP_i^{ue}$ denotes the non-event-driven FLOPs, mainly membrane-potential accumulation and the first linear layer. We adopt the standard 45nm proxy with $E_{AC}=0.9$ pJ and $E_{MAC}=4.6$ pJ from "Computing's Energy Problem (and what we can do about it)". We will add this calculation protocol explicitly in the revision.
>
> ---
>
> **Question 3.** The energy benefit is not due to lambda alone. Lambda controls the amplitude of the position-dependent threshold modulation in PE-LIF, but the reduction in energy comes from the SPE mechanism itself: PE-LIF changes the threshold distribution of the Q/K neurons, suppresses firing activity on average, and therefore reduces downstream attention activity. If lambda were zero, SPE would collapse to a non-positional threshold setting and lose its main effect. Thus, the observed energy gain is a consequence of the SPE design together with its threshold parameterization, not an isolated hyperparameter trick.
>
> ---
>
> We will thoroughly revise the manuscript according to your comments, and we express our sincere gratitude once more.

---

### Official Review · Reviewer_BuZG · 2026-03-10

**Soundness:** 3
**Presentation:** 3
**Significance:** 3
**Originality:** 3
**Overall Recommendation:** 4
**Confidence:** 4

**Summary:**

This paper proposes a positional encoding method for Spiking Transformers by encoding positional information into neuron thresholds. In this way, positional information can be implicitly embedded into the output spikes during the firing process of spiking neurons, yielding a positional encoding scheme that is compatible with spike-driven computation. The method is theoretically argued to represent relative positional information while preserving the linear attention property. In addition, the paper introduces MPR-SG as an additional training mechanism to facilitate the proposed method. The effectiveness of the overall method is validated on multiple NLP datasets.

**Compliance With Llm Reviewing Policy:**

Affirmed.

**Final Justification:**

The rebuttal addressed my main concerns well and reinforced my overall positive assessment of the paper. I will maintain my original score.

**Key Questions For Authors:**

Please clarify how the reported energy consumption is computed. In particular, is it based on a hardware-independent proxy (e.g., firing-rate / AC-MAC cost estimation) or measured on a specific platform? A more explicit description of the calculation protocol would make the energy-efficiency claim easier to assess.

**Limitations:**

The paper mainly validates the proposed positional encoding through downstream task performance, and more direct evaluation of positional modeling ability is still missing.

**Strengths And Weaknesses:**

### Strengths

1. The proposed threshold-based positional encoding differs from conventional positional encoding methods in ANNs, which typically act directly on input representations or attention scores. It is better aligned with the spike-based representation and computational mechanism of SNNs.
2. Without introducing additional parameters, the proposed method shows overall performance gains across multiple evaluation tasks, especially on long-text benchmarks and several order-sensitive tasks.
3. The paper provides fairly detailed mathematical derivations and theoretical analysis, and further supports the proposed method with ablation studies.

### Weaknesses

1. The effectiveness of the positional encoding is mainly validated through downstream task performance, which is still an indirect form of evaluation. The paper lacks more direct experimental evidence on the positional modeling ability itself.
2. Although the paper reports energy evaluation results and provides a qualitative explanation for the reduction in energy consumption, it does not clearly present the methodology or formulas used for the energy analysis.
3. The theoretical justification of the method relies on the assumption that membrane potential statistics are approximately consistent with firing-rate statistics. This assumption is not a natural property of neuron dynamics and instead depends on an additional training mechanism to be approximately satisfied. As a result, it remains unclear how reliably the method would generalize to a broader range of architectures and training paradigms.

---

> ### Author Rebuttal · Authors · 2026-03-31
>
> # Response to Reviewer BuZG
>
> We thank the reviewer for the insightful questions on analysis and evaluation.
>
> ---
>
> **Response to Weakness 1.** We agree that downstream accuracy is an indirect evaluation of positional modeling. However, in the current paper, there are near direct evidences which comes from two sources: (i) the theoretical analysis, where Proposition 4.1 shows that PE-LIF introduces a term depending on relative distance and Proposition 4.2 establishes long-term decay; and (ii) the empirical pattern that the largest gains appear on tasks known to depend strongly on positional/structural information, such as CoLA (+8.9), STS-B (+24.8), AGNEWS, and IMDB. We agree that dedicated probing experiments would strengthen the paper, but we cannot add a rigorous new probe suite during the rebuttal period. We will clarify this limitation and add a more explicit discussion of positional-modeling evidence in the revision.
>
> ---
>
> **Response to Weakness 2 / Question 1.** We agree that the energy methodology should be stated explicitly. Table 2 reports analytical, hardware-independent estimates rather than measurements on a specific platform. For layer $l$, we estimate the number of spike operations as
> $$SOP_l = fr \times T \times FLOP_l^e$$
> where $fr$ is the firing rate, $T$ is the number of time steps, and $FLOP_l^e$ denotes the FLOPs of the event-driven part. The total energy is then estimated by
> $$E = E_{AC}\sum_{i=1}^L SOP_i + E_{MAC}\sum_{i=1}^L FLOP_i^{ue}$$
> where $FLOP_i^{ue}$ denotes the FLOPs of the non-event-driven part, mainly membrane-potential accumulation and the first linear layer. Following the standard 45nm proxy in [1], we use $E_{AC}=0.9$ pJ and $E_{MAC}=4.6$ pJ. We will add this protocol, together with its assumptions and interpretation, to the revised paper.
>
> ---
>
> **Response to Weakness 3.** We agree that the condition relating membrane-potential statistics and firing-rate statistics is not an automatic property of arbitrary neuron dynamics. For this reason, the paper does not simply assume it for free; instead, we introduce MPR-SG to encourage the condition to hold approximately in the relevant PE-LIF layers. Its necessity is supported by the ablation results (64.00 -> 64.68 -> 65.97 average on GLUE dev) and by Fig. 4 / Appendix C, where MPR-SG consistently reduces the gap between membrane-potential statistics and firing-rate statistics. We will soften the wording to emphasize approximate validity in the studied LIF-based setup, rather than universal generalization across arbitrary architectures or training paradigms.
>
> ---
>
>  We will thoroughly revise the manuscript according to your comments, and we express our sincere gratitude once more.
>
> ---
>
> [1] Computing's Energy Problem (and what we can do about it)

---

> > ### Author Rebuttal · Reviewer_BuZG · 2026-04-03
> >
> > Thank you for the detailed rebuttal. The authors have satisfactorily addressed my concerns, and I do not have further questions.

---

> > > ### Author Response · Authors · 2026-04-07
> > >
> > > Dear Reviewer BuZG,
> > >
> > > We are glad that your concerns have been addressed. We sincerely appreciate your recognition of our work. We will revise the manuscript carefully based on your suggestions, as well as those from the other reviewers, to further improve its quality.
> > >
> > > Best regards,\
> > > The Authors

---

### Official Review · Reviewer_4Kw5 · 2026-03-12

**Soundness:** 3
**Presentation:** 3
**Significance:** 2
**Originality:** 2
**Overall Recommendation:** 4
**Confidence:** 4

**Summary:**

This paper studies positional encoding for Spiking Transformers and proposes Spiking Positional Encoding (SPE), which injects positional information into position-dependent firing thresholds via PE-LIF neurons instead of adding it to embeddings or attention scores. The paper further introduces MPR-SG to better support the theoretical condition underlying the method. Experiments on six text classification datasets and GLUE show consistent improvements over prior SNN positional encoding methods and the baseline Spikingformer.

**Compliance With Llm Reviewing Policy:**

Affirmed.

**Final Justification:**

I raise the score to 4.

**Key Questions For Authors:**

Please refer to the issues raised in the Weaknesses section above.

**Limitations:**

yes

**Strengths And Weaknesses:**

### Strengths

1. The paper is generally well written and easy to follow. The motivation, method, and experimental sections are clearly organized.

2. Positional encoding is indeed a central missing component for Spiking Transformers in sequence tasks, and the paper identifies a real gap between ANN-style PE and the constraints of spike-driven computation.

3.  Beyond introducing the method itself, the paper also provides problem analysis and theoretical arguments to justify why the design can capture relative positional information and long-term decay behavior. Although some assumptions are strong, the overall theory-method connection is fairly clear.

4. The paper makes a clearer effort to justify the role of MPR-SG, both through performance ablations and through membrane-potential distribution plots. This helps connect the training method to the theoretical condition it is meant to support.

### Weaknesses

1. While the paper adapts positional encoding to the spiking setting in a reasonable way, the core positional form itself appears relatively conventional. In particular, the position-dependent threshold design on top of LIF neurons is still largely based on a sinusoidal-style encoding scheme, which has been widely studied in prior work. Therefore, the contribution seems closer to a spiking-specific adaptation of an existing idea than to a genuinely new positional encoding design.

2. The proposed method seems most naturally applicable to directly trained Spiking Transformers. However, current SNN LLM research is largely dominated by conversion-based approaches, which are typically much stronger than directly trained spiking models in both scale and accuracy. As a result, the practical scope of the proposed method may currently be narrower than the paper suggests.

3. Although the method is motivated in part by the SNN setting, the paper does not sufficiently discuss whether the proposed position-dependent threshold mechanism is compatible with existing neuromorphic hardware. This is particularly important because a major advantage of SNNs lies in their efficient hardware implementation on neuromorphic platforms. It is therefore unclear how easily PE-LIF could be implemented in practice, and whether the proposed design would preserve the hardware-level benefits typically associated with standard spiking neurons.

4. The current version does not clearly discuss where the method may fail, such as sensitivity to sequence length, dependence on architecture choice, transfer to other domains beyond NLP, or the cost of relying on a specific theoretical assumption enforced through training.

5. While the paper positions SPE as a general positional encoding method for Spiking Transformers, the empirical evaluation is conducted on a relatively narrow set of baselines. This limits the extent to which the experiments support the broader claims of general effectiveness.

---

> ### Author Rebuttal · Authors · 2026-03-31
>
> # Response to Reviewer 4Kw5
>
> We thank the reviewer for the careful assessment and for highlighting several scope-related issues.
>
> ---
>
> **Weakness 1.** We agree that the trigonometric basis itself is not the main novelty. The novelty lies in the *spiking-native realization*: SPE injects positional information through position-dependent firing thresholds in PE-LIF neurons, so that positional cues are carried by spike generation and membrane reset dynamics rather than by adding continuous embeddings to activations. This design is specifically motivated by the constraints of Spiking Transformers, and our theoretical analysis shows that it can encode relative positional information while preserving the linear-attention property of SSA. We will revise the paper to emphasize that the contribution is a spiking-specific mechanism and analysis, not a new sinusoidal family.
>
> ---
>
> **Weakness 2.** We agree that the current paper is most directly applicable to *directly trained* Spiking Transformers, and we will narrow the claim accordingly. This is precisely the regime where positional information must be incorporated without breaking spike-driven computation and linear SSA. We do not claim that the present work resolves positional encoding for conversion-based SNN LLMs, whose constraints are different. Studying how threshold-based positional encoding interacts with ANN-to-SNN conversion is an important direction, but it is beyond the scope of the current submission.
>
> ---
>
> **Weakness 3.** We agree that the deployment aspect should be discussed more concretely. Current SNN deployment research primarily focuses on FPGAs, whose development workflow is also an important reference for neuromorphic-chip design and validation. From this perspective, the threshold matrix required by SPE can be stored in off-chip RAM. When a processing element (PE) needs the threshold at a specific position, the value is first transferred from off-chip RAM to on-chip RAM and then read by the PE. This off-chip-to-on-chip transfer is serialized, but its latency is small: assuming a 250 MHz off-chip RAM clock and 16-bit data width, transmitting a 128 x 768 threshold matrix of 16-bit values takes approximately 196.608 us, which is negligible. The subsequent on-chip RAM-to-PE access can proceed in parallel with weight fetching via multiple read channels, so it introduces no additional latency. In summary, threshold-matrix access is highly similar to standard weight access and does not introduce extra hardware complexity, resource overhead, or access latency. We will add this discussion to the revision to clarify practical deployability.
>
> ---
>
> **Weakness 4.** We agree that both the transfer scope and the failure modes should be stated more clearly. Regarding transfer beyond NLP, we have now conducted additional vision experiments and observe consistent gains: on DVS-CIFAR10, accuracy improves from 81.3 to 81.5, and on ImageNet from 66.86 to 67.63 using Spikingformer-8-256. This suggests that SPE can generalize beyond NLP, although the gain is smaller in vision, likely because convolutional front-ends already provide part of the positional bias. At the same time, limitations remain: in our NLP experiments, SPE underperforms the baseline on small/simple datasets such as RTE and MRPC, which we attribute to overfitting; moreover, the current validation is still centered on LIF-based Spikingformer-style architectures, and MPR-SG only enforces the required condition approximately. We will make these boundaries explicit in the paper.
>
> |Method|Acc.(DVS-CIFAR10)|Acc.(ImageNet)|
> |-|-|-|
> |Spikingformer|81.3|66.86|
> |Spikingformer w/ SPE|**81.5**|**67.63**|
>
> ---
>
> **Weakness 5.** We agree that the claim of generality should be phrased carefully; however, we would also like to clarify that the empirical evidence is broader than a single narrow comparison. First, Table 1 compares SPE against multiple representative SNN positional encoding methods (CPG-PE, Gray-PE, Log-PE) across six text-classification benchmarks spanning both short and long sequences. Second, Table 2 provides a matched-backbone evaluation on GLUE, which isolates the effect of SPE under the same Spikingformer architecture. Third, with the additional experiments above, we also observe consistent gains on vision tasks (DVS-CIFAR10 and ImageNet). Taken together, these results support the claim that SPE is a generally useful *drop-in positional encoding strategy* rather than a gain tied to a single dataset. We agree that the current evidence is still concentrated on the Spikingformer family and does not yet establish universality across all Spiking Transformer variants. We will therefore benchmark SPE on additional spiking backbones beyond Spikingformer in our future work.
>
> ---
>
> We will thoroughly revise the manuscript according to your comments, and we express our sincere gratitude once more.

---

> > ### Author Rebuttal · Reviewer_4Kw5 · 2026-04-02
> >
> > The authors solved the concern of the reviewer.

---

> > > ### Author Response · Authors · 2026-04-07
> > >
> > > Dear Reviewer 4Kw5,
> > >
> > > We are glad that your concerns have been addressed. We sincerely appreciate your recognition of our work. We will revise the manuscript carefully based on your suggestions, as well as those from the other reviewers, to further improve its quality.
> > >
> > > Best regards,\
> > > The Authors

---

### Official Review · Reviewer_xk9o · 2026-03-13

**Soundness:** 2
**Presentation:** 2
**Significance:** 2
**Originality:** 2
**Overall Recommendation:** 4
**Confidence:** 3

**Summary:**

The paper proposes Spiking Positional Encoding (SPE), a positional encoding mechanism tailored to Spiking Transformers for sequential NLP tasks. The central idea is to encode position information not by adding embeddings to tokens or modifying attention scores directly, but by introducing a PE-LIF neuron layer whose firing thresholds vary by position and dimension. The authors argue that this preserves spike-driven computation and linear attention while allowing relative positional information to be implicitly represented in the spike activations. They further introduce MPR-SG, a surrogate-gradient modification intended to encourage the condition needed by their relative-position proposition.

**Compliance With Llm Reviewing Policy:**

Affirmed.

**Final Justification:**

Thanks authors for detailed explanation, I overlooked the image resolution used in the experiments is small, and there is no text (multi modal input), which is quite different from the current Vision-Language models use. I raised my score to 4.

**Key Questions For Authors:**

N/A

**Limitations:**

Yes

**Strengths And Weaknesses:**

[Strength]

1. Transformer-based SNNs are becoming stronger, but positional encoding for sequence tasks remains underexplored, and ANN positional encoding methods are argued to be mismatched with spike-driven linear-attention architectures.

2. On six text classification datasets, SPE outperforms prior SNN positional encoding baselines such as CPG-PE, Gray-PE, and Log-PE, and on GLUE it improves the Spikingformer baseline by 5.8 average points.


[Weakness]

1.  The related work positioning is somewhat overstated. The paper says positional encoding in Spiking Transformers is “largely unexplored,” which is directionally fair, but it also cites recent SNN-specific works such as CPG-PE, Gray-PE, and Log-PE. Given that, the true novelty is not introducing positional encoding to SNNs per se, but introducing a threshold-based method that aims to preserve linear attention while representing relative position. I think the paper should state that more explicitly.

2. The method is only validated on NLP-style sequence tasks, despite broader Spiking Transformer framing. Since the related work section includes many vision Spiking Transformers and argues that convolution can serve as implicit positional encoding. Could the authors provide the results on vision task?

3. Although the paper includes multiple SNN positional encoding baselines on the text-classification benchmarks, the main GLUE result is essentially a comparison to one matched baseline (Spikingformer without SPE). I would have liked to see more fully matched comparisons against alternative positional encoding methods under the same backbone and training setup.

---

> ### Author Rebuttal · Authors · 2026-03-31
>
> # Response to Reviewer xk9o
>
> We thank the reviewer for the constructive suggestions.
>
> ---
>
> **Weakness 1.** We agree that the novelty should be positioned more precisely. Our contribution is not the first use of positional cues in SNNs per se. Rather, SPE is a spiking-specific, threshold-based positional encoding for sequence-oriented Spiking Transformers that simultaneously (i) preserves spike-driven computation, (ii) encodes relative positional information, and (iii) remains compatible with linear SSA. As discussed in Section 2.2, prior SNN positional encodings such as CPG-PE, Gray-PE, and Log-PE do not satisfy this combination: they either do not model relative position or are not designed to preserve the linear-attention setting. We will revise the wording accordingly and replace the broader "largely unexplored" phrasing with a narrower claim.
>
> ---
>
> **Weakness 2.** We thank the reviewer for this suggestion. We have now conducted additional vision experiments with the Spikingformer backbone. SPE improves Spikingformer from 81.3 to 81.5 on DVS-CIFAR10, and from 66.86 to 67.63 on ImageNet under the Spikingformer-8-256 architecture. These results show that SPE is not limited to NLP and can also provide gains for vision Spiking Transformers. At the same time, the improvement is more moderate than on sequence tasks, which is consistent with our discussion that convolutional front-ends in vision models may already introduce an implicit positional bias. We will add these results and revise the paper to distinguish the strongest evidence in sequence modeling from the broader, but still positive, transfer to vision.
>
> |Method|Acc.(DVS-CIFAR10)|Acc.(ImageNet)|
> |-|-|-|
> |Spikingformer|81.3|66.86|
> |Spikingformer w/ SPE|**81.5**|**67.63**|
>
> ---
>
> **Weakness 3.** We agree that stronger matched GLUE baselines would further strengthen the paper. In the current submission, Table 2 isolates the drop-in effect of SPE under the same Spikingformer backbone and training recipe, while Table 1 compares SPE against multiple SNN positional encoding baselines on text-classification benchmarks. A fully matched GLUE comparison against CPG-PE/Gray-PE/Log-PE would require re-implementing or re-pretraining those alternatives under the same language-model backbone and optimization pipeline, which is beyond the rebuttal window. We will clarify this limitation and tone down the empirical claim to effectiveness in the evaluated Spikingformer setting rather than universal superiority.
>
> ---
>
> We will thoroughly revise the manuscript according to your comments, and we express our sincere gratitude once more.

---

> > ### Author Rebuttal · Reviewer_xk9o · 2026-04-03
> >
> > Thanks for the response. I have carefully reviewed the response. Although my first/third concern has been resolved, Weakness 2 still shows that the proposed method achieves marginal improvement in the vision domain. Could the authors provide an explanation of why the vision domain achieves relatively less improvement than the NLP domain?

---

> > > ### Author Response · Authors · 2026-04-06
> > >
> > > Once again, thank you for your invaluable contribution to improving the quality of our manuscript. Should our response adequately address your concerns, we would be most grateful if you would reconsider your evaluation score.
> > >
> > > ## Preliminary
> > >
> > > Positional encoding schemes designed for NLP tasks cannot be directly applied to vision tasks. In NLP, tokens form a one-dimensional sequence, whereas in vision, tokens are arranged on a two-dimensional grid. If this two-dimensional arrangement is simply flattened row by row into a one-dimensional sequence and encoded with a one-dimensional positional encoding scheme, a critical problem arises: tokens that are spatially adjacent along the column dimension may become distant in the flattened sequence, separated by at least one full row length.
> > >
> > > To accommodate the two-dimensional structure inherent in vision tasks, we extend our method to the two-dimensional case as follows:
> > > $$
> > > \theta_{x,y,k} =
> > > \begin{cases}
> > > \theta + \lambda \cdot \cos\left(\frac{x}{10000^{\frac{k-1}{D}}}\right) & x=1,2,\ldots,N_x, k=1,5,\ldots,D-3, \\\\
> > > \theta + \lambda \cdot \sin\left(\frac{x}{10000^{\frac{k-2}{D}}}\right) & x=1,2,\ldots,N_x, k=2,6,\ldots,D-2, \\\\
> > > \theta + \lambda \cdot \cos\left(\frac{y}{10000^{\frac{k-3}{D}}}\right) & y=1,2,\ldots,N_y, k=3,7,\ldots,D-1, \\\\
> > > \theta + \lambda \cdot \sin\left(\frac{y}{10000^{\frac{k-4}{D}}}\right) & y=1,2,\ldots,N_y, k=4,8,\ldots,D.
> > > \end{cases}
> > > $$
> > >
> > > Here, $x$ and $y$ denote the horizontal and vertical coordinates in the image, respectively. As we show below, 2D PE-LIF is capable of encoding two-dimensional relative positional information.
> > >
> > > > **Proposition 3** *When employing the 2D PE-LIF neuron layer to obtain activations*
> > > >
> > > > $$
> > > > \begin{cases}
> > > > Q(t) = [\boldsymbol{q} _{1,1}(t), \boldsymbol{q} _{1,2}(t), \ldots, \boldsymbol{q} _{N_x,N_y}(t)]^\top \in {0,1}^{N_x N_y \times D}, \\\\
> > > > K(t) = [\boldsymbol{k} _{1,1}(t), \boldsymbol{k} _{1,2}(t), \ldots, \boldsymbol{k} _{N_x,N_y}(t)]^\top \in {0,1}^{N_x N_y \times D},
> > > > \end{cases}
> > > > $$
> > > >
> > > > *under the condition $\mathbb{E}\left[\hat{u} _{x,y,k}(t)\right] = \mathbb{E}\left[s(t)\right]$, there exists a function $g$ that takes $x_1 - x_2$ and $y_1 - y_2$ as its arguments such that $\mathbb{E}\left[\boldsymbol{q} _{x_1,y_1}^\top(t) \boldsymbol{k} _{x_2,y_2}(t)\right]$ contains $g(x_1 - x_2, y_1 - y_2)$, where*
> > > >
> > > > $$
> > > > \begin{aligned}
> > > > g(x_1 - x_2, y_1 - y_2) = &\frac{1}{2} \sum_{k=1}^{D/4} \left(\mathbb{E}\left[\mathcal{B} _{4k-3,4k-3}^{(x_1,x_2,y_1,y_2)}\right] + \mathbb{E}\left[\mathcal{B} _{4k-2,4k-2}^{(x_1,x_2,y_1,y_2)}\right]\right) \cos(x_1 - x_2)\beta_k \\\\
> > > > &+ \frac{1}{2} \sum _{k=1}^{D/4} \left(\mathbb{E}\left[\mathcal{B} _{4k-1,4k-1}^{(x_1,x_2,y_1,y_2)}\right] + \mathbb{E}\left[\mathcal{B} _{4k,4k}^{(x_1,x_2,y_1,y_2)}\right]\right) \cos(y_1 - y_2)\beta_k.
> > > > \end{aligned}
> > > > $$
> > > >
> > > > *This shows that 2D PE-LIF can directly encode relative positional information into $Q(t)$ and $K(t)$.*
> > >
> > > ## Why Is the Improvement in the Vision Domain Smaller Than That in the NLP Domain?
> > >
> > > In the NLP domain, the token sequence length is 128 in most experiments, and in some cases it can reach 1024. Positional encoding can play a noticeable role in such long-sequence tasks. For example, the largest performance gains are observed on the two tasks with a sequence length of 1024.
> > >
> > > In contrast, for the ImageNet classification task in the vision domain, the token grid is of size $14 \times 14$, meaning that the sequence length along each spatial dimension is only 14—far shorter than 128 or 1024. In such a setting, the impact of positional encoding is inherently limited. For the DVS-CIFAR10 dataset, the token grid shrinks further to $8 \times 8$, making the effect even less pronounced.

---

### Decision · Program_Chairs · 2026-04-30

**Decision:**

Accept (regular)

**Comment:**

The paper proposes the Spiking Positional Encoding (SPE) method, which provides an innovative and effective solution to the positional encoding problem in Spiking Transformers. Through the position-dependent threshold design of the PE-LIF neuron layer, it enables implicit encoding of relative positional information while preserving spike-driven computation and linear attention.Experimental results show that the method significantly outperforms existing SNN positional encoding baselines across multiple NLP text classification datasets and the GLUE benchmark, and also achieves consistent improvements when extended to vision tasks, with solid theoretical analysis and experimental validation.The authors have provided detailed and reasonable responses in the rebuttal to the concerns raised by the reviewers regarding novelty positioning, experimental scope, hardware compatibility, and other issues, effectively addressing most of them and supplementing key experimental data and explanations.Although there is still room for improvement in terms of the magnitude of gains on vision tasks and the completeness of ablation studies, the overall work is technically sound and makes a clear contribution that can advance the subfield of Spiking Transformers. We therefore recommend acceptance.